# A hierarchical decomposition for explaining ML performance discrepancies

**Harvineet Singh**[1]    **Fan Xia**[1]    **Adarsh Subbaswamy**[2]    **Alexej Gossmann**[2]    **Jean Feng**[1*]

[1]University of California, San Francisco
[2]U.S. Food and Drug Administration, Center for Devices and Radiological Health

## Abstract

Machine learning (ML) algorithms can often differ in performance across domains. Understanding *why* their performance differs is crucial for determining what types of interventions (e.g., algorithmic or operational) are most effective at closing the performance gaps. *Aggregate decompositions* express the total performance gap as the gap due to a shift in the feature distribution $p(X)$ plus the gap due to a shift in the outcome's conditional distribution $p(Y|X)$. While this coarse explanation is helpful for guiding root cause analyses, it provides limited details and can only suggest coarse fixes involving **all variables** in an ML system. *Detailed decompositions* quantify the importance of **each variable** to each term in the aggregate decomposition, which can provide a deeper understanding and suggest more targeted interventions. Although parametric methods exist for conducting a full hierarchical decomposition of an algorithm's performance gap at the aggregate and detailed levels, current nonparametric methods only cover parts of the hierarchy; many also require knowledge of the entire causal graph. We introduce a nonparametric hierarchical framework for explaining why the performance of an ML algorithm differs across domains, without requiring causal knowledge. Furthermore, we derive debiased, computationally-efficient estimators and statistical inference procedures to construct confidence intervals for the explanations.

## 1   Introduction

The performance of an ML algorithm can differ across domains due to shifts in the data distribution. Understanding what contributed to this performance gap can help teams choose the most effective corrective action(s), ranging from algorithmic modifications (e.g. model retraining) to operational fixes (e.g. updating data pipelines). Prior works have focused primarily on *aggregate* decompositions, which decompose the performance gap into that due to a shift in the marginal distribution of the input features $p(X)$ (covariate shift [37]) and that due to a shift in the conditional distribution of the outcome $p(Y|X)$ (concept shift or conditional outcome shift) [5, 50, 30, 36, 16]. However, coarse decompositions can only suggest coarse corrective actions, such as investigating data pipelines for all features. The goal of this work is to provide a hierarchical nonparametric framework that first decomposes a performance gap into aggregate terms and then each aggregate term into detailed terms. This helps narrow down the features to investigate and understand how they affect the gap.

If one is willing to make the strong assumption that the expected loss of a model is a linear function of some feature set $X$, the problem of obtaining *aggregate* and *detailed* decompositions drastically simplifies. This is the key assumption underlying the Oaxaca-Blinder (OB) decomposition, one of the most widely used frameworks in the (income and health) disparities literature [32, 3]. Given an ML

---

*Corresponding author: `jean.feng@ucsf.edu`

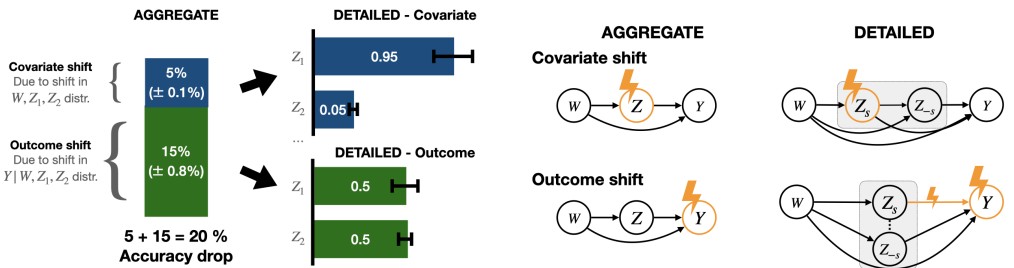

Figure 1: (**left**) Proposed framework called Hierarchical Decomposition of Performance Differences (namely, HDPD) helps to understand performance gaps of an ML algorithm between two domains. It decomposes the overall gap (say, in classification accuracy) into gaps due to shifts in the covariate versus outcome distribution (Aggregate). Then, it quantifies the importance of each feature to the two components (Detailed). (**right**) In terms of directed acyclic graphs, aggregate decompositions describe the effect of shift interventions, for instance, on the outcome $Y$ distribution while keeping all else fixed between domains. Detailed decompositions quantify how well can we explain those shift interventions by more targeted shifts with respect to feature subsets $Z_s$ alone.

model with average loss $\mathbb{E}_D[\ell]$ in domains $D = 0$ and $1$ and assuming the linear loss relationship $\mathbb{E}_D[\ell|X] = \beta_D^\top X$, the OB framework decomposes the performance gap at the *aggregate* level into that due to a covariate shift $(\beta_0^\top (\mathbb{E}_1[X] - \mathbb{E}_0[X]))$ and that due to a conditional outcome shift $((\beta_1 - \beta_0)^\top \mathbb{E}_1[X])$. That is, the former is due to a shift in the feature means and the latter is due to a shift in the coefficients. At the *detailed* level, the aggregate terms corresponding to covariate and conditional outcome shifts are further broken down into the contributions from each feature, i.e. $\beta_{0,j}(\mathbb{E}_1[X_j] - \mathbb{E}_0[X_j])$ and $(\beta_{1,j} - \beta_{0,j})\mathbb{E}_1[X_j]$, respectively. Although the highly intuitive nature of the OB framework has led to its widespread popularity, the terms are difficult to interpret under model misspecification. As such, this work aims to define a similar hierarchical decomposition framework for explaining ML performance disparities, *without making strong parametric assumptions*.

There is currently no unified, nonparametric framework that obtains aggregate and detailed decompositions. Instead, solutions have been proposed for parts of the hierarchy (see Table 1): nonparametric methods exist for the aggregate decomposition [5, 30, 48] and, assuming the causal graph is known, detailed decompositions of the covariate shift [40, 44, 50, 38, 4, 23]. However, the causal graph is unlikely to be known in high-dimensional settings and, more importantly, there are no methods for simultaneously obtaining a detailed decomposition of the conditional outcome shift. There are also methods that do not decompose the performance gap and instead describe distribution shifts in the variables [29] or model explanations [12, 31, 23]. However, such approaches do not quantify how such shifts ultimately contribute to an ML performance gap. We make the following contributions.

- We introduce a **unified hierarchical nonparametric framework** for decomposing the performance gap of an ML algorithm (Fig 1 left). Using the concept of *partial* distribution shifts, we generalize shifts with respect to variable subgroups to encompass not only covariate shifts but also conditional outcome shifts. We then introduce a unified scoring rule for (candidate) partial shifts, which can be used even when the causal graph is not known.
- We derive novel debiased and asymptotically normal estimators for terms in the decomposition, which allow us to construct **confidence intervals (CIs) with asymptotically valid coverage rates**.
- We demonstrate the utility of our framework in **real-world examples** of prediction models for hospital readmission and insurance coverage. Code for reproducing experiments is available at `https://github.com/jjfeng/HDPD`.

## 2 A unifying framework for explaining performance gaps

**Notation.** Consider a prediction algorithm $f : \mathcal{X} \subseteq \mathbb{R}^m \to [0, 1]$ for binary outcomes $Y$ across source and target domains, denoted by $D = 0$ and $D = 1$, respectively. Let the performance of $f$ be quantified in terms of a loss function $\ell : \mathcal{X} \times \{0, 1\} \to \mathbb{R}$, such as the 0-1 misclassification loss $\mathbb{1}\{f(X) \neq Y\}$. Suppose variables $X$ can be partitioned into disjoint sets $W \in \mathbb{R}^{m_1}$ and $Z = X \setminus W \in \mathbb{R}^{m_2}$, where $m = m_1 + m_2$. Although our framework does not require knowing the causal ordering between variables, the interpretation is more intuitive when $W$ is causally upstream

Table 1: Comparison of HDPD to prior works that decompose ML performance gaps. The distinguishing contribution of this work is that it unifies aggregate and detailed decompositions under a nonparametric framework with uncertainty quantification.

| Papers | Aggregate decomp. | Detailed decomp. for | | Does not require causal graph | Confidence intervals | Nonparametric |
|---|---|---|---|---|---|---|
| | | $p(X)$-shift | $p(Y\|X)$-shift | | | |
| Zhang et al. [50] | ✓ | ✓ | | | | ✓ |
| Cai et al. [5] | ✓ | | | ✓ | ✓ | ✓ |
| Quintas-Martinez et al. [38] | ✓ | ✓ | | | ✓ | ✓ |
| Wu et al. [48] | | ✓ | | ✓ | | ✓ |
| Liu et al. [30] | ✓ | | ✓ | ✓ | | ✓ |
| Dodd and Pepe [11] | | | ✓ | ✓ | ✓ | |
| Oaxaca [32], Blinder [3] | ✓ | ✓ | ✓ | ✓ | ✓ | |
| HDPD (this paper) | ✓ | ✓ | ✓ | ✓ | ✓ | ✓ |

of $Z$ and $Y$ (Fig 1 right). Variables $Z$ can be chosen to be mediators or modifiers of the effect of the domain shift $D$ on $Y$. For instance, if $Z$ are treatment variables and $W$ are baseline variables, one can interpret a covariate shift as a change in the treatment policy and an outcome shift as a change in the treatment effect across the two environments. In absence of any causal knowledge, another option is to choose $W$ as the variables for which one would like the expected loss given $W$ to be invariant across the two environments; this can be useful to promote fairness of ML algorithms across environments. When this invariance does not hold, the framework explains how variables $Z$ contribute to these differences. We refer to $W$ as baseline variables and $Z$ as conditional covariates. Please refer to Appendix C for more discussion on choosing $W$ and $Z$ as well as a summary of notation (Table 2).

Our proposed hierarchical decomposition of an ML performance gap is based on a stratification of distribution shifts into *aggregate* and *partial* shifts. At the aggregate level, the joint distribution of $(W, Z, Y)$ can be factorized with respect to the aggregate variable groups $W, Z$, and $Y$, i.e.

$$p_{D_W}(W)p_{D_Z}(Z|W)p_{D_Y}(Y|W, Z), \tag{1}$$

where subscripts $D_W, D_Z$ and $D_Y$ indicate the domain of that factor. An *aggregate shift* substitutes a factor from the source domain in (1) with that from the target domain, i.e. we swap the factor from $p_0$ to $p_1$. A *partial shift* with respect to variable subset $s$ (or an $s$-partial shift) shifts a factor from the source domain in (1) *only with respect to variable subset $s$*; we denote this by swapping a factor from $p_0$ to $p_s$. (We keep the precise definition of $s$-partial shifts purposely vague until Section 2.2.) We denote expectations with respect to the joint distribution (1) as $\mathbb{E}_{D_W D_Z D_Y}$.

The overall performance gap between domains, $\Lambda = \mathbb{E}_{111}\left[\ell(W, Z, Y)\right] - \mathbb{E}_{000}\left[\ell(W, Z, Y)\right]$, can be decomposed hierarchically as follows. In the Appendix D, we also discuss how this decomposition can be interpreted causally under certain conditions.

**Aggregate.** At the first level of the hierarchy, the framework quantifies how aggregate shifts contribute to the performance gap individually. This leads to the decomposition $\Lambda = \Lambda_W + \Lambda_Z + \Lambda_Y$, where $\Lambda_W$ quantifies the impact of a shift in the baseline distribution $p(W)$, $\Lambda_Z$ quantifies the impact of a shift in the conditional covariate distribution $p(Z|W)$, and $\Lambda_Y$ quantifies the impact of a shift in the outcome distribution $p(Y|W, Z)$. More concretely,

$$\Lambda_W = \mathbb{E}_{100}\left[\ell\right] - \mathbb{E}_{000}\left[\ell\right]$$
$$\Lambda_Z = \mathbb{E}_{110}[\ell] - \mathbb{E}_{100}[\ell] = \mathbb{E}_{1\cdot\cdot}\Big[\underbrace{\mathbb{E}_{\cdot10}\left[\ell \mid W\right] - \mathbb{E}_{\cdot00}\left[\ell \mid W\right]}_{\Delta_{\cdot10}(W)}\Big]$$
$$\Lambda_Y = \mathbb{E}_{111}[\ell] - \mathbb{E}_{110}[\ell] = \mathbb{E}_{11\cdot}\Big[\underbrace{\mathbb{E}_{\cdot\cdot1}\left[\ell \mid W, Z\right] - \mathbb{E}_{\cdot\cdot0}\left[\ell \mid W, Z\right]}_{\Delta_{\cdot\cdot0}(W,Z)}\Big].$$

The same (or similar) aggregate decompositions have also appeared in prior works [5, 30, 15, 50, 38].

**Detailed.** At the detailed level, each aggregate term is further broken down into variable-level attributions. The effect of each variable can be isolated using partial shifts. However, because variables can interact to induce complex partial distribution shifts, we define variable importance (VI) using the Shapley attribution framework [41, 6, 31, 17], which has the benefits of satisfying axiomatic properties such as fairness, monotonicity, and full attribution. Thus, given a real-valued value function $v$ that quantifies the contribution of an $s$-partial shift to an aggregate shift for all

$s \subseteq \{1, \cdots, m\}$, the attribution to variable $j$ is the average gain in value when additionally shifting with respect to $j$, i.e.

$$\phi_j := \frac{1}{m} \sum_{s \subseteq \{1, \cdots, m\} \backslash j} \binom{m-1}{|s|}^{-1} \{v(s \cup j) - v(s)\}. \tag{2}$$

**Interpretation.** Such VI values can help ML teams identify the underlying cause(s) for a performance gap and design targeted operational and/or algorithmic interventions. For instance, a variable with high importance to the conditional covariate shift term $\Lambda_Z$ may indicate differences in the variable's missingness rates, prevalence, or selection bias across domains. If instead the variable is highly important to the conditional outcome shift term $\Lambda_Y$, it may indicate inherent differences in the conditional distribution (i.e. effect modification), differences in measurement error or the way outcome is defined between domains, or omission of variables predictive of the outcome. Finally, note that variable importances should be viewed as *relative* to the variables included in the framework rather than absolute importances, as one cannot include all possible explanatory variables.

To define VI values, **the key question is how to define a value function $v$ that is applicable to different types of $s$-partial shifts, even when the causal graph is not known.** It turns out that the answer is far from straightforward. The next section discusses how the value function and candidate $s$-partial shifts must be defined with care.

## 2.1 Value of partial distribution shifts

When the true causal graph is known, prior works define an $s$-partial covariate shift as the substitution of nodes $s$ with mechanisms from the target domain and its value $v(s)$ as the difference in the average loss, e.g. $\mathbb{E}_{1s0}[\ell] - \mathbb{E}_{100}[\ell]$ [50, 38]. However, this has a number of limitations: (i) knowing the entire causal graph is often impractical, (ii) in the absence of such a graph, this value function is not a proper scoring rule and can assign high values to partial shifts that contradict the true causal graph (see Example E.1 for details), and (iii) $v(s)$ can be high even if the shift does not induce similar shifts in the loss as the aggregate shift.

Instead, we propose to evaluate candidate $s$-partial shifts by how closely they approximate aggregate shifts, using a nonparametric extension of the traditional $R^2$ measure. In the case of conditional covariate shifts, an aggregate shift induces a performance difference of $\Delta_{\cdot 10}(W)$ in strata $W$ while a candidate $s$-partial shift induces a performance difference of $\Delta_{\cdot s0}(W) = \mathbb{E}_{\cdot s0}[\ell|W] - \mathbb{E}_{\cdot 00}[\ell|W]$. The value of this $s$-partial shift is then the percent variation of $\Delta_{\cdot 10}$ explained by $\Delta_{\cdot s0}$, i.e.

$$v_Z(s) := 1 - \frac{\mathbb{E}_{1\cdot\cdot}\left[(\Delta_{\cdot s0}(W) - \Delta_{\cdot 10}(W))^2\right]}{\mathbb{E}_{1\cdot\cdot}[\Delta_{\cdot 10}^2(W)]}. \tag{3}$$

Likewise, for conditional outcome shifts, an aggregate shift induces a performance difference of $\Delta_{\cdot\cdot 1}(W, Z)$ in strata $(W, Z)$ while a candidate $s$-partial shift induces a performance difference of $\Delta_{\cdot\cdot s}(W, Z) = \mathbb{E}_{\cdot\cdot s}[\ell|W, Z] - \mathbb{E}_{\cdot\cdot 0}[\ell|W, Z]$. The value of this $s$-partial conditional outcome shift is then defined as the percent variation of $\Delta_{\cdot\cdot 1}$ explained by $\Delta_{\cdot\cdot s}$, i.e.

$$v_Y(s) := 1 - \frac{\mathbb{E}_{11\cdot}\left[(\Delta_{\cdot\cdot s}(W, Z) - \Delta_{\cdot\cdot 1}(W, Z))^2\right]}{\mathbb{E}_{11\cdot}[\Delta_{\cdot\cdot 1}^2(W, Z)]}. \tag{4}$$

This formulation of the value function in terms of $R^2$ provides a unified way to score partial conditional covariate and outcome shifts, does not require knowledge of the true causal graph, and is a strictly proper scoring rule under certain conditions (see Appendix E). In general, we expect the highest scoring candidate $s$-partial shifts to be those that are close to the true causal graph *and* induce large shifts in the ML algorithm's loss. Finally, we acknowledge one caveat with this framework: because some variables must be held out to define the $R^2$ measure, we cannot score partial shifts in the baseline variables $W$. We hope to close this gap in future work.

## 2.2 Candidate partial distribution shifts

We now present the set of candidate partial shifts considered in this work. High-level illustrations for the candidate partial shifts are given in Fig 1 right top; more detailed illustrations are given in

Fig 4 of the Appendix. We emphasize that these are *candidates*, as the true causal graph is not known. While there are certainly other partial shifts that one may consider, many have various disadvantages. As such, we leave the investigation of other partial shifts to future work.

$s$**-partial conditional covariate shift**: Suppose $Z_{-s}$ is downstream of $Z_s$. Then $p_s(z|w) := p_1(z_s|w)p_0(z_{-s}|z_s, w)$. Wu et al. [48] considered a similar proposal.

$s$**-partial conditional outcome shift**: Shifting the conditional distribution of $Y$ only with respect to a variable subset $Z_s$ but not $Z_{-s}$ requires care. We cannot simply define $p_s(Y|W, Z)$ as a function of only $W$ and $Z_s$. Such a definition would imply that an $s$-partial shift has a non-zero effect, even in settings with no shift in the conditional outcome distribution (i.e. $p_1(Y|W, Z) \equiv p_0(Y|W, Z)$).

Instead, we define an $s$-partial outcome shift based on models commonly used in model recalibration/revision [42, 34], where the modified risk (conditional probability of $Y$) is a function of the risk in the source domain $Q := q(W, Z) := p_0(Y = 1|W, Z)$, $W$, and $Z_s$. That is, we define the shift as

$$p_s(y|z, r, w) := p_1(y|z_s, r, w) = \int p_1(y|\tilde{z}_{-s}, z_s, w)p_1(\tilde{z}_{-s}|z_s, q(w, z_s, \tilde{z}_{-s}) = r, w)d\tilde{z}_{-s} \quad (5)$$

By defining the shifted outcome distribution solely as a function of $Q, W$, and $Z_s$, any direct effect from $Z_{-s}$ to $Y$ is eliminated and $p_s$ has the desired behavior in the setting where there is no conditional outcome shift.

# 3 Estimation and statistical inference

Here we discuss estimation and statistical inference for the aggregate terms ($\Lambda_{\mathtt{W}}$, $\Lambda_{\mathtt{Z}}$, and $\Lambda_{\mathtt{Y}}$), the value functions $v_{\mathtt{Z}}(s)$ and $v_{\mathtt{Y}}(s)$, and the Shapley-based detailed terms $\phi_{\mathtt{Z},j}$ and $\phi_{\mathtt{Y},j}$ for $j \in (0, \cdots, m_2)$. One approach is to rely on *plug-in* estimators, which plug in estimates of conditional means (also called outcome models) or density ratios [43], which we collectively refer to as nuisance parameters. For instance, one can estimate the conditional means $\mu_{\cdot 10}(w) = \mathbb{E}_{\cdot 10}[\ell|W]$ and $\mu_{\cdot 00}(w) = \mathbb{E}_{\cdot 00}[\ell|W]$ using ML and take the empirical mean of $\hat{\mu}_{\cdot 10} - \hat{\mu}_{\cdot 00}$ with respect to the target domain to get a plug-in estimator for $\Lambda_{\mathtt{Z}} = \mathbb{E}_{1 \cdot \cdot}[\mu_{\cdot 10} - \mu_{\cdot 00}]$. However, because estimation of the true nuisance parameters using ML typically converge at a rate slower than $n^{-1/2}$, plug-in estimators generally fail to be consistent at a rate of $n^{-1/2}$ and cannot be used to construct CIs [25].

To this end, we use the method of one-step correction from semiparametric inference to derive *debiased* ML estimators [45, 7]. The core idea is to subtract the first-order bias of a plug-in estimator, which requires characterizing the canonical gradient (or efficient influence function) of the estimand [25]. The primary technical contribution in this section is the derivation of debiased estimators for the detailed decompositions. (Estimation and inference for the aggregate decomposition is well-studied, as the aggregate terms can be formulated as average causal effects.) Due to space limitations, this section only presents estimators for the detailed decomposition of the conditional outcome shift. This estimand is particularly interesting, as its unique structure is not amenable to standard techniques for debiasing ML estimators. We refer the reader to the Appendix for derivations, pseudocode, and proofs for all the estimators.

**Notation.** Let $\mathbb{P}_D$ denote the expectation with respect to domain $D$. For ease of exposition, suppose the number of IID observations from each domain is the same, denoted by $n$. We present split-sample estimators, though the results can be readily extended using cross-fitting [7, 25]. Let the data be randomly split into "training" and "evaluation" partitions. Let $\mathbb{P}_{D,n}$ denote the empirical average in the evaluation partition for domain $D$. All estimated quantities are denoted using hat notation.

## 3.1 Value of $s$-partial conditional outcome shifts

Here we describe the high-level steps for deriving a debiased estimator for $v_{\mathtt{Y}}(s)$, the value of a candidate $s$-partial conditional outcome shift. The following section describes a computationally efficient procedure for combining such estimates to obtain Shapley values.

Standard recipes for deriving asymptotically normal, nonparametric-efficient estimators rely on *pathwise* differentiability of the estimand and analyzing its efficient influence function [25]. However, $v_{\mathtt{Y}}(s)$ is not pathwise differentiable because it is a function of (5), which conditions on the source risk $q(w, z)$ equalling some value $r$. Taking the pathwise derivative of $v_{\mathtt{Y}}(s)$ requires taking a derivative

of the indicator function $\mathbb{1}\{q(w,z)=r\}$, which generally does not exist. Given the difficulties in deriving an asymptotically normal estimator for $v_Y(s)$, we propose estimating a close alternative that *is* pathwise differentiable.

The idea is to replace $q$ in (5) with its binned variant $q_{\mathtt{bin}}(w,z) = \frac{1}{B}\lfloor q(w,z)B + \frac{1}{2}\rfloor$ for some $B \in \mathbb{Z}^+$, which discretizes outputs from $q$ into $B$ disjoint bins. As long as $B$ is sufficiently high, the binned version of the estimand, denoted $v_{Y,\mathtt{bin}}(s)$, is a close approximation to $v_Y(s)$. (We use $B = 20$ in the empirical analyses, which we believe to be sufficient in practice.) The benefit of this binned variant is that the derivative of the indicator function $\mathbb{1}\{q_{\mathtt{bin}}(w,z)=r\}$ is zero almost everywhere as long as observations with source risks exactly equal to a bin edge have measure zero. More formally, we require the following:

**Condition 3.1.** *Let $\Xi$ be the set of $(W,Z)$ such that $q(W,Z)$ falls precisely on some bin edge and is not equal to zero or one. The set $\Xi$ is measure zero.*

Under this condition, $v_{Y,\mathtt{bin}}(s)$ is pathwise differentiable and, using one-step correction, we derive a debiased ML estimator that has the unique form of a V-statistic (this follows from the integration over "phantom" $\tilde{z}_{-s}$ in (5)). We represent V-statistics using the operator $\mathbb{P}_{1,n}\tilde{\mathbb{P}}_{1,n}$, which takes the average over all pairs of observations $O_i$ with replacement, i.e. $\frac{1}{n^2}\sum_{i=1}^{n}\sum_{j=1}^{n}g(O_i,O_j)$ for some function $g$. Calculation of this estimator and its theoretical properties are as follows.

**Estimation.** Using the training partition, estimate the outcome models $\mu_{..D}(W,Z) = \mathbb{E}_{..D}[\ell|W,Z]$ for $D = 0,1$, the shifted outcome model $\mu_{..s}(W,Z) = \mathbb{E}_{..s}[\ell|W,Z]$; and the density ratio models $\pi_{110}(W,Z) = p_1(W,Z)/p_0(W,Z)$ and $\pi(W,Z_s,Z_{-s},Q_{\mathtt{bin}}) = p_1(Z_{-s}|W,Z_s,q_{\mathtt{bin}}(W,Z) = Q_{\mathtt{bin}})/p_1(Z_{-s})$. The outcome and density ratio models can be fit using ML-based regression models and probabilistic classifiers [43], respectively (see Section H for details). The estimator for $v_{Y,\mathtt{bin}}(s)$ is the ratio $\hat{v}_{Y,\mathtt{bin}}(s) = \hat{v}_{Y,n}^{\mathtt{num}}(s)/\hat{v}_{Y,n}^{\mathtt{den}}$, where the numerator and denominator are estimated using the evaluation partition as

$$\begin{aligned}
\hat{v}_{Y,n}^{\mathtt{num}}(s) = {} & \mathbb{P}_{1,n}\hat{\xi}_s(W,Z)^2 + 2\,\mathbb{P}_{1,n}\hat{\xi}_s(W,Z)(\ell - \hat{\mu}_{..1}(W,Z)) \\
& - 2\,\mathbb{P}_{1,n}\tilde{\mathbb{P}}_{1,n}\hat{\xi}_s(W,Z_s,\tilde{Z}_{-s})\ell(W,Z_s,\tilde{Z}_{-s},Y)\hat{\pi}(W,Z_s,\tilde{Z}_{-s},Q_{\mathtt{bin}}) \\
& + 2\,\mathbb{P}_{1,n}\tilde{\mathbb{P}}_{1,n}\hat{\xi}_s(W,Z_s,\tilde{Z}_{-s})\hat{\mu}_{..s}(W,Z_s,\tilde{Z}_{-s})\hat{\pi}(W,Z_s,\tilde{Z}_{-s},Q_{\mathtt{bin}}) \quad (6) \\
\hat{v}_{Y,n}^{\mathtt{den}} = {} & \mathbb{P}_{1,n}\left(\hat{\mu}_{..1}(W,Z) - \hat{\mu}_{..0}(W,Z)\right)^2 + 2\,\mathbb{P}_{1,n}\left(\hat{\mu}_{..1}(W,Z) - \hat{\mu}_{..0}(W,Z)\right)(\ell - \hat{\mu}_{..1}(W,Z)) \\
& - 2\,\mathbb{P}_{0,n}\left(\hat{\mu}_{..1}(W,Z) - \hat{\mu}_{..0}(W,Z)\right)(\ell - \hat{\mu}_{..0}(W,Z))\hat{\pi}_{110}(W,Z), \quad (7)
\end{aligned}$$

where $\hat{\xi}_s(W,Z) = \hat{\mu}_{..1}(W,Z) - \hat{\mu}_{..s}(W,Z)$. Note that the first terms in (6) and (7) are the plug-in estimates, followed by additional terms that correct its bias.

**Inference.** This estimator is asymptotically normal assuming the estimators for the nuisance parameters converge at a fast enough rate, per the following theorem.

**Theorem 3.2.** *Suppose Condition 3.1 holds. For variable subset $s$, suppose the density ratios $\pi(W,Z_s,Z_{-s},Q_{bin})$ and $\pi_{110}(W,Z)$ are bounded; denominator in case of no shift $v_Y^{den}(\emptyset) > 0$; estimator $\hat{\pi}$ is consistent; estimators $\hat{\mu}_{..0}, \hat{\mu}_{..1}$ and $\hat{\mu}_{..s}$ converge at an $o_p(n^{-1/4})$ rate, and*

$$\mathbb{P}_1(\hat{q}_{bin} - q_{bin})^2 = o_p(n^{-1}) \quad (8)$$

$$\mathbb{P}_1(\mu_{..s} - \hat{\mu}_{..s})(\pi - \hat{\pi}) = o_p(n^{-1/2}), \quad \mathbb{P}_0(\mu_{..0} - \hat{\mu}_{..0})(\pi_{110} - \hat{\pi}_{110}) = o_p(n^{-1/2}) \quad (9)$$

*Then the estimator $\hat{v}_{Y,bin}(s)$ is asymptotically normal centered at the estimand $v_{Y,bin}(s)$.*

Note that the product terms in (9) mean that the estimator converges to normal at $n^{-1/2}$-rate even if one of the nuisance parameters is estimated at a rate slower than $n^{-1/2}$. Hence, it is *multiply-robust* to nuisance model misspecification. A convergence rate of $o_p(n^{-1/4})$ can be achieved by ML estimators in a wide variety of conditions, and such assumptions are commonly used to construct debiased ML estimators. The additional requirement in (8) that $\hat{q}_{\mathtt{bin}}$ converges at a $o_p(n^{-1})$ rate is new, but fast or even super-fast convergence rates of *binned* risks is achievable under suitable margin conditions [2] such as Condition G.7 in the Appendix.

## 3.2 Shapley values

Calculating the exact Shapley value is computationally intractable as it involves an exponential number of terms. However, Williamson and Feng [47] showed that calculating the exact Shapley

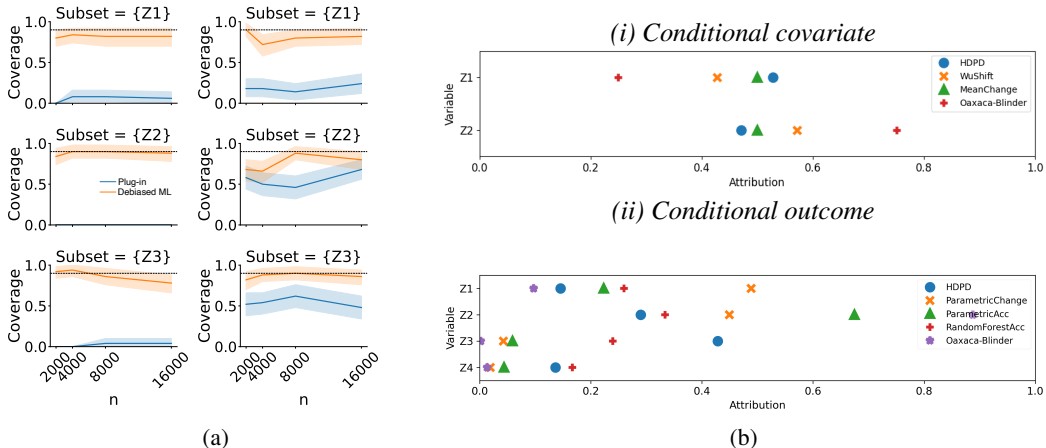

Figure 2: (a) Coverage rates of 90% CIs for value of $s$-partial shifts for the conditional covariate (first column) and outcome shifts (second column) across dataset sizes $n$. Dashed horizontal line indicates 90% coverage rate. (b) Comparison of variable importance reported by proposed method HDPD (debiased) versus existing methods for conditional covariate and outcome shift terms.

value is unnecessary for the purposes of statistical inference. Because there is inherent uncertainty in estimates of the value functions $v(s)$, one only needs to sample and estimate the values for enough variable subsets such that the uncertainty due to estimation dominates that due to subset sampling. This leads to a drastic reduction in computation time: Williamson and Feng [47] proves that the number of subsets one needs to sample only needs to be linear or super-linear in the total number of observations $n$. Using this result, Algorithm 4 outlines a computationally efficient procedure for estimation and inference of the detailed decomposition.

## 4 Simulation

We now present simulations to show that the proposed procedure achieves the desired coverage rates (Section 4.1) and illustrate how the HDPD framework provides more intuitive explanations of performance gaps (Section 4.2). In all empirical analyses, performance of the ML algorithm is quantified in terms of 0-1 accuracy. Below, we briefly describe the simulation settings; full details are provided in Section I in the Appendix.

### 4.1 Verifying theoretical properties

We first verify that the inference procedures for the decomposition terms have CIs with coverage close to their nominal rate. We check the coverage of the aggregate decomposition as well as the value of $s$-partial conditional covariate and partial conditional outcome shifts for $s = \{Z_1\}, \{Z_2\}, \{Z_3\}$. $(W, Z_1, Z_2, Z_3)$ are sampled from independent normal distributions with different means in source and target, while $Y$ is simulated from logistic regression models with different coefficients. CIs for the debiased ML estimator converge to the nominal 90% coverage rate with increasing sample size, whereas those for the naïve plug-in estimator do not (Fig 2a and Fig 6).

### 4.2 Comparing explanations

We now compare the proposed definitions for the detailed decomposition with existing methods. For the detailed decomposition due to conditional covariate shift, the comparators are:

- `MeanChange` Tests for a difference in means for each feature. Defines importance as $1-$ p-value.
- `Oaxaca-Blinder`: Fits a linear model of the logit-transformed expected loss with respect to $Z$ in the source domain. Defines importance of $Z_i$ as its coefficient multiplied by the difference in the means of $Z_i$ [32, 3].
- `WuShift` [48]: Defines importance of subset $s$ as change in *overall* performance due to $s$-partial conditional covariate shifts. Applies Shapley framework to obtain VIs.

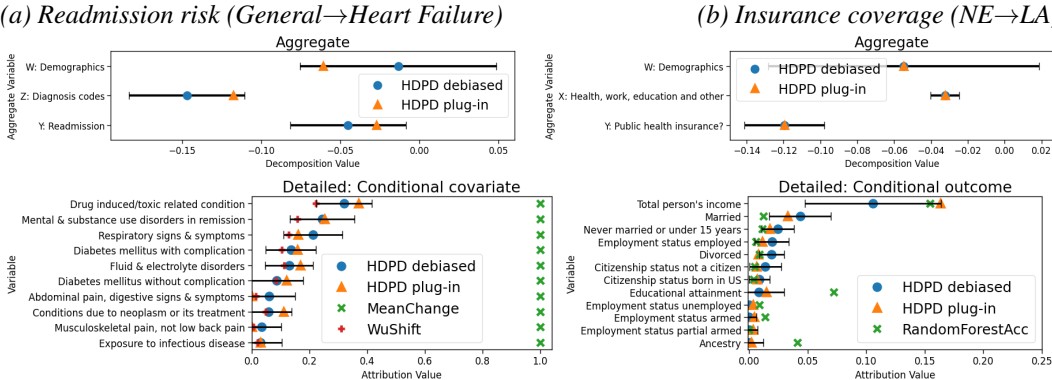

Figure 3: Aggregate and detailed decompositions for performance gaps of (a) a model predicting readmission risk across patient populations and (b) a model predicting insurance coverage across US states. A subset of VI estimates is shown; see full list in Section J in the Appendix.

For the detailed decomposition due to conditional outcome shifts, we compare against:

- `ParametricChange`: Fits a logistic model for $Y$ with interaction terms between domain and $Z$. Defines importance of $Z_i$ as the coefficient of its interaction term.
- `ParametricAcc`: Same as `ParametricChange` but models the 0-1 loss rather than $Y$.
- `RandomForestAcc`: Compares VI of random forest models trained on data from both domains with input features $D$, $Z$, and $W$ to predict the 0-1 loss.
- `Oaxaca-Blinder`: Fits linear models for the logit-transformed expected loss in each domain. Defines importance of $Z_i$ as its mean in the target domain multiplied by the difference in its coefficients across domains.

Although the proposed method may agree with these other methods on the top features in certain data settings, we highlight important situations where the methods differ.

**Conditional covariate.** (Fig 2b(i)) We simulate $(W, Z_1)$ from a standard normal distribution, $Z_2$ from a mixture of two Gaussians whose means depend on the value of $Z_1$ (i.e. $Z_1 \rightarrow Z_2$), and $Y$ from a logistic regression model depending on $(W, Z_1, Z_2)$. We induce a shift from the source domain to the target domain by shifting only the distribution of $Z_1$, so that $p_1(Z|W) = p_0(Z_2|Z_1, W)p_1(Z_1|W)$. Only the proposed estimator correctly recovers that $Z_1$ is more important than $Z_2$, as the $\{1\}$-partial conditional covariate shift explains all the variation in performance gaps across strata $W$ (i.e. the corresponding $R^2$-based value function $v_Z(\{1\})$ is equal to 1). The other methods incorrectly assign higher importance to $Z_2$. `MeanChange` only measures shifts but not loss due to shifts, `Oaxaca-Blinder` uses a misspecified linear model, and `WuShift` estimates the performance change due to hypothesized $s$-partial shifts but does not check if the partial shifts are good explanations in the first place.

**Conditional outcome.** (Fig 2b(ii)) $W$ and $Z \in \mathbb{R}^4$ are simulated from the same distribution in both domains. $Y$ is generated from a logistic regression model with coefficients for $(W, Z_1, \cdots, Z_4)$ as $(0.5, 0.5, 1, 0.3, 0.3)$ in the source and $(0.5, 0.3, 1, 1.3, -0.1)$ in the target. Interestingly, none of the methods have the same ranking of the features. `ParametricChange` identifies $Z_1$ as having the largest shift on the logit scale, but this does not mean that it is the most important explanation for changes in the *loss*. According to our decomposition framework, $Z_3$ is actually the most important for explaining changes in model performance due to outcome shifts. `Oaxaca-Blinder`, `ParametricAcc`, and `RandomForestAcc` have odd behavior. `Oaxaca-Blinder` assigns $Z_3$ second to the lowest importance and `ParametricAcc` assigns $Z_2$ the highest importance), likely because they misspecify the outcome models. `RandomForestAcc` likely ranks $Z_2$ highly because its VI values quantify which variables are good predictors of performance, not performance shift.

A more objective evaluation is to compare the performance of fixes based on the different explanations. To this end, we re-fit the ML algorithm in the target domain with respect to input features $Q, W$, and the top variables $Z_s$ from each explanation. We find that model revisions based on the proposed method achieve the highest performance gain (Table 3 in Appendix).

# 5 Real-world data case studies

We now demonstrate applicability of the framework on two datasets with naturally-occurring shifts.

**Hospital readmission.** Using electronic health record data from the Zuckerberg San Francisco General Hospital, we analyzed performance of a Gradient Boosted Tree (GBT) trained on the general patient population (source) to predict 30-day readmission risk but applied to patients diagnosed with heart failure (HF, target). Features include 4 demographic variables ($W$) and 16 diagnosis codes ($Z$). Each domain supplied $n = 3750$ observations from which we keep 20% in the evaluation partition.

Model accuracy drops from 70% to 53% in HF population. From the aggregate decompositions (Fig 3a), we observe that the drop is mainly due to covariate shift. If one performed the standard check to see which variables significantly changed in their mean value (`MeanChange`), then one would find a significant shift in nearly *every* variable. Little support is offered to identify main drivers of the performance drop. In contrast, the detailed decomposition from the proposed framework estimates diagnoses "Drug-induced or toxic-related condition" and "Mental & substance use disorder in remission" as having the highest estimated contributions to the conditional covariate shift, and most other variables having little to no contribution. Upon discussion with clinicians from this hospital, differences in the top two diagnoses may be explained by (i) substance use being a major cause of HF at this hospital, with over eighty percent of its HF patients reporting current or prior substance use, and (ii) substance use and mental health disorders often occurring simultaneously in this HF patient population. Based on these findings, closing the performance gap may require a mixture of both operational (e.g. care programs centered around substance use) and algorithmic interventions (e.g. reweighting data with respect to the top two features). Finally, CIs from the debiased ML procedure provide valuable information on the uncertainty of the estimates and highlight, for instance, that more data is necessary to determine the true ordering between the top two features. In contrast, existing methods do not provide (asymptotically valid) CIs.

**ACS Public Coverage.** We analyze a neural network trained to predict whether a person has public health insurance using data from Nebraska in the American Community Survey (source, $n = 3000$), applied to data from Louisiana (target, $n = 6000$). Baseline variables include 3 demographics (sex, age, race), and covariates $Z$ include 31 variables related to health conditions, employment, marital status, citizenship status, and education.

Model accuracy drops from 84% to 66% across the two states. The main driver is the shift in the outcome distribution per the aggregate decomposition (Fig 3b) and the most important contributor to the outcome shift is annual income, perhaps due to differences in cost of living across the two states. Income is significantly more important than all the other variables; the ranking between the remaining variables is unclear. In comparing the performance of targeted model revisions, we find that revising the model based on top variables identified by the proposed procedure leads to AUCs that are better or as good as those based on `RandomForestAcc` (Table 4 in the Appendix).

# 6 Prior work

**Describing distribution shifts.** This line of work focuses on detecting and localizing which distributions shift between datasets [29, 39]. Budhathoki et al. [4] identify the main variables contributing to a distribution shift via a Shapley framework, Kulinski and Inouye [28] fits interpretable optimal transport maps, and Liu et al. [30] finds the region with the largest shift in the conditional outcome distribution. However, these works do not quantify how these shifts contribute to *changes in performance*, the metric of practical importance.

**Explaining loss differences across subpopulations.** Understanding differences in model performance across subpopulations in a single dataset is similar to understanding differences in model performance across datasets, but the focus is typically to *find* subpopulations with poor performance rather than to explain how distribution shifts contributed to the performance change. Existing approaches include slice discovery methods [35, 22, 10, 13] and structured representations of the subpopulation using e.g. Euclidean balls [1].

**Attributing performance changes.** Prior works have described similar aggregate decompositions of the performance change into covariate and conditional outcome shift components [5, 36]. To provide more granular explanations of performance shifts, existing works on causal attribution [50, 38] and

mediation analysis [44] quantify the importance of shifts in each variable assuming the causal graph is correctly specified; covariate shifts restricted to variable subsets assuming that the partial shifts follow a particular structure [48]; and conditional shifts in each variable assuming a parametric model [11]. However, the strong assumptions made by these methods make them difficult to apply in practice, and model misspecification can lead to unintuitive interpretations. Furthermore, such methods do not provide hierarchical decompositions, i.e. VIs for each type of shift. Decomposition methods such as Oaxaca-Blinder similarly make strong parametric assumptions [32, 3, 16, 14, 49, 15], which is inappropriate for the complex data settings in ML. In addition, there is no unifying nonparametric framework for decomposing both covariate and outcome shifts, and many methods do not output CIs, which is important when the amount of labeled data from a given domain is limited. A summary of how the proposed framework compares against prior works is shown in Table 1.

## 7   Discussion

ML algorithms regularly encounter distribution shifts in practice, leading to drops in performance. We present a novel framework that helps ML developers and deployment teams build a more nuanced understanding of the shifts. Compared to past work, the approach provides a nonparametric hierarchical framework for decomposing both conditional covariate and outcome shifts, does not require fine-grained knowledge of the causal relationship between variables, and quantifies the uncertainty of the estimates by constructing confidence intervals. We present real-world case studies to demonstrate how this framework can help diagnose performance drops and guide corrective actions. This framework requires overlapping support of the covariates, which may not always be applicable in practice. In such cases, one solution is to restrict to the common support [5].

Important extensions of this work include decompositions of more complex measures of model performance such as AUC and analyzing other factorizations of the data distribution (e.g. label/prior shifts [27]). For unstructured data (e.g. image and text), the current framework can be applied to low-dimensional embeddings or by extracting interpretable concepts [26]; more work is needed to directly analyze unstructured data. Finally, while the focus of this work is to interpret performance gaps, future work may take this work one step further to design optimal interventions for closing the performance gap.

## Acknowledgments and Disclosure of Funding

We would like to thank Lucas Zier for supplying the dataset from the Zuckerberg San Francisco General Hospital and providing their clinical expertise to interpret results. We are grateful to Nicholas Petrick, Berkman Sahiner, Gene Pennello, Mi-Ok Kim, Romain Pirracchio, Julian Hong, Avni Kothari, and the anonymous reviewers for helpful feedback. This work was funded through a Patient-Centered Outcomes Research Institute® (PCORI®) Award (ME-2022C1-25619). The views presented in this work are solely the responsibility of the author(s) and do not necessarily represent the views of the PCORI®, its Board of Governors or Methodology Committee. The contents are those of the author(s) and do not necessarily represent the official views of, nor an endorsement, by FDA/HHS, or the U.S. Government.

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

# A    Appendix

Contents of the Appendix are as follows.

- Table 2 collects all the notation used for reference.

- Section B discusses broader impacts of the work.

- Algorithms 1 and 4 provide the steps required for computing the aggregate and detailed decomposition respectively. Detailed decompositions require computing the value of $s$-partial conditional outcome and conditional covariate shifts which is described in Algorithms 2 and 3.

- Section C discusses considerations for choosing baseline variables $W$ and conditional covariates $Z$.

- Section D describes a causal interpretation of the aggregate and detailed decompositions as effects of stochastic (or shift) interventions on a structural causal model.

- Section E explain why value functions in prior work give unintuitive attribution and that the $R^2$-based value functions are proper scoring rules.

- Section F describes the estimation and inference for aggregate decomposition and detailed decomposition of conditional covariate shift.

- Section G provides the derivations of the results.

- Sections H and I describe the implementation and simulation details.

- Section J provides additional details on the two real world datasets and results.

# B    Broader impacts

This work presents a method for understanding failures of ML algorithms when they are deployed in settings or populations different from the ones in development datasets. Therefore, the work can be used to suggest ways of improving the algorithms or mitigating their harms. The method is generally applicable to tabular data settings for any classification algorithm, hence, it can potentially be applied across multiple domains where ML is used including medicine, finance, and online commerce.

Care must be taken while interpreting the results. As usual, assumptions underlying the decompositions such as the coarse causal ordering between the variables $W, Z$, and $Y$ should be validated through domain knowledge.

---

**Algorithm 1** Aggregate decompositions into baseline, conditional covariate, and conditional outcome shifts

---

**Input:** Source and target data $\{(W_i^{(d)}, Z_i^{(d)}, Y_i^{(d)})\}_{i=1}^{n_d}$ for $d \in \{0, 1\}$, loss function $\ell(W, Z, Y; f)$.
**Output:** Performance change due to baseline, conditional covariate, and conditional outcome shifts $\Lambda_{\mathtt{W}}, \Lambda_{\mathtt{Z}}, \Lambda_{\mathtt{Y}}$.

1   Split source and target data into training $\mathtt{Tr}$ and evaluation $\mathtt{Ev}$ partitions. Let $n^{\mathtt{Ev}}$ be the total number of data points in the $\mathtt{Ev}$ partition.

2   Fit nuisance parameters $\eta_{\mathtt{W}}, \eta_{\mathtt{Z}}, \eta_{\mathtt{Y}}$, defined in Section G.1, on the $\mathtt{Tr}$ partition as outlined in Section H.1.

3   Estimate $\Lambda_{\mathtt{W}}, \Lambda_{\mathtt{Z}}, \Lambda_{\mathtt{Y}}$ using fitted nuisance parameters on the $\mathtt{Ev}$ partitions following the equations in Section F.1.

4   Estimate variance of influence functions $\psi_{\mathtt{W}}(d, w, z, y; \hat{\eta}_{\mathtt{N}}), \psi_{\mathtt{Z}}(d, w, z, y; \hat{\eta}_{\mathtt{N}})$, and $\psi_{\mathtt{Y}}(d, w, z, y; \hat{\eta}_{\mathtt{N}})$ as defined in (23), (24), and (25), respectively.

5   Compute $\alpha$-level confidence intervals as $\hat{\Lambda}_{\mathtt{N}} \pm z_{1-\alpha/2} \sqrt{\widehat{var}(\psi_{\mathtt{N}}(d, w, z, y; \hat{\eta}_{\mathtt{N}}))/n^{\mathtt{Ev}}}$ for $\mathtt{N} \in \{\mathtt{W}, \mathtt{Z}, \mathtt{Y}\}$, where $z$ is the inverse CDF of the standard normal distribution.

6   **return** $\hat{\Lambda}_{\mathtt{W}}, \hat{\Lambda}_{\mathtt{Z}}, \hat{\Lambda}_{\mathtt{Y}}$ *and confidence intervals*

---

Table 2: Notation

| Symbol | Meaning |
|--------|---------|
| $W, Z, Y$ | Variables: Baseline, Conditional covariates, Outcome |
| $f$ | Prediction model being analyzed |
| $\ell(W, Z, Y)$ or $\ell$ | Loss function e.g. 0-1 loss |
| $D = 0$ and $D = 1$ | Indicators for source and target domain |
| $p_0, p_1$ | Probability density (or mass) function for the two domains $D = 0, 1$ |
| $p_s$ | Probability density (or mass) function when only variable subset $s$ shifts from source to target |
| $\mathbb{E}_{D_W D_Z D_Y}$ | Expectation over the distribution $p_{D_W}(W) p_{D_Z}(Z\|W) p_{D_Y}(Y\|W, Z)$ |
| $Q := q(W, Z)$ | Source domain risk at $W, Z$, i.e. $p_0(Y = 1\|W, Z)$ |
| $\mathtt{Tr}$ and $\mathtt{Ev}$ | Training dataset used to fit models and evaluation dataset used to compute decompositions |
| $\phi_{Z,j}$ and $\phi_{Y,j}$ | Shapley values for variable $j$ in the detailed decomposition of conditional covariate and outcome shifts |
| $v_Z(s)$ and $v_Y(s)$ | Value of a subset $s$ for $s$-partial conditional covariate shift and $s$-partial outcome shift |
| $v_\cdot^{\mathtt{num}}(s)$ and $v_\cdot^{\mathtt{den}}(s)$ | Numerator and denominator of the ratio defined in the value of a subset |
| Models $\mu_\cdot$ | Outcome models for the conditional expectation of the loss across different settings |
| Models $\pi_\cdot$ | Density ratio models for feature densities across datasets |
| $\mathbb{P}$ | Notation for expectation |
| $\mathbb{P}_{0,n}$ and $\mathbb{P}_{1,n}$ | sample average over source and target data in the evaluation dataset |
| $\psi(d, w, z, y)$ | Influence function defined in the linear approximation of an estimand, see e.g. (22) |

---

**Algorithm 2** VALUECONDITIONALOUTCOME(S): Value for $s$-partial conditional outcome shift for a subset $s$

---

**Input:** Training $\mathtt{Tr}$ and evaluation $\mathtt{Ev}$ partitions of source and target data, subset of variables $s$.
**Output:** Value for $s$-partial conditional outcome shift for subset $s$.

1 Fit nuisance parameters $\eta_{Y,s}^{\mathtt{num}}, \eta_Y^{\mathtt{den}}$, defined in Sections G.3, on the $\mathtt{Tr}$ partitions as outlined in H.2.

2 Estimate $v_Y(s)$ by $\hat{v}_Y^{\mathtt{num}}(s)/\hat{v}_Y^{\mathtt{den}}$ where $\hat{v}_Y^{\mathtt{num}}(s)$ is estimated using (6) and $\hat{v}_Y^{\mathtt{den}}$ is estimated using (7) on the $\mathtt{Ev}$ partition.

3 Estimate variance of influence function $\psi_{Y,\mathtt{bin},s}(d, w, z, y; \hat{\eta}_{Y,s}^{\mathtt{num}}, \hat{\eta}_Y^{\mathtt{den}})$ as defined in (74).

4 Compute $\alpha$-level confidence interval as $\hat{v}_Y(s) \pm z_{1-\alpha/2}\sqrt{\widehat{var}(\psi_{Y,\mathtt{bin},s}(d, w, z, y; \hat{\eta}_{Y,s}^{\mathtt{num}}, \hat{\eta}_Y^{\mathtt{den}}))/n^{\mathtt{Ev}}}$.

5 **return** $\hat{v}_Y(s)$ *and confidence interval*

---

**Algorithm 3** VALUECONDITIONALCOVARIATE(S): Value for $s$-partial conditional covariate shift for a subset $s$

---

**Input:** Training $\mathtt{Tr}$ and evaluation $\mathtt{Ev}$ partitions of source and target data, subset of variables $s$.
**Output:** Value for $s$-partial conditional covariate shift for subset $s$.

1 Fit nuisance parameters $\eta_{Z,s}^{\mathtt{num}}$, defined in Sections G.2, on the $\mathtt{Tr}$ partition, as outlined in H.3.

2 Estimate $v_Z(s)$ by $\hat{v}_Z^{\mathtt{num}}(s)/\hat{v}_Z^{\mathtt{num}}(\emptyset)$ using (21) on the $\mathtt{Ev}$ partition.

3 Estimate variance of influence function $\psi_{Z,s}(d, w, z, y; \hat{\eta}_{Z,s}^{\mathtt{num}})$ as defined in (41).

4 Compute $\alpha$-level confidence interval as $\hat{v}_Z(s) \pm z_{1-\alpha/2}\sqrt{\widehat{var}(\psi_{Z,s}(d, w, z, y; \hat{\eta}_{Z,s}^{\mathtt{num}}))/n^{\mathtt{Ev}}}$.

5 **return** $\hat{v}_Z(s)$ *and confidence interval*

---
**Algorithm 4** Detailed decomposition for conditional outcome and covariate shift
---
**Input:** Source and target data $\{(W_i^{(d)}, Z_i^{(d)}, Y_i^{(d)})\}_{i=1}^{n_d}$ for $d \in \{0, 1\}$, loss function $\ell(W, Z, Y; f)$, $\gamma \in \mathbb{R}^+$.

**Output:** Detailed decomposition for conditional outcome or covariate shift, $\{\phi_{\mathtt{Y},j} : j = 0, \cdots, m_2\}$ or $\{\phi_{\mathtt{Z},j} : j = 1, \cdots, m_2\}$.

1 Split source and target data into training $\mathtt{Tr}$ and evaluation $\mathtt{Ev}$ partitions. Let $n^{\mathtt{Ev}}$ be the total number of data points in the $\mathtt{Ev}$ partition.

2 Subsample $\lfloor \gamma n^{\mathtt{Ev}} \rfloor$ subsets from $\mathcal{Z} = \{1, \cdots, m_2\}$ with respect to Shapley weights, including $\emptyset$ and $\mathcal{Z}$, denoted $s_1, \cdots, s_k$.

3 Estimate $v_{\mathtt{Y}}(s) \leftarrow$ VALUECONDITIONALOUTCOME(S) and $v_{\mathtt{Z}}(s) \leftarrow$ VALUECONDITIONALCO-VARIATE(S) for $s \in s_1, \cdots, s_k$.

4 Get estimated Shapley values $\{\phi_{\mathtt{Y},j}\}$ and $\{\phi_{\mathtt{Z},j}\}$ by solving constrained linear regression problems in (7) in Williamson and Feng [47] with value functions $v_{\mathtt{Y}}(s)$ and $v_{\mathtt{Z}}(s)$, respectively.

5 Compute confidence intervals based on the influence functions defined in Theorem 1 in Williamson and Feng [47].

6 **return** *Shapley values* $\{\phi_{\mathtt{Y},j} : j = 0, \ldots, m_2\}$ *and* $\{\phi_{\mathtt{Z},j} : j = 1, \ldots, m_2\}$ *and confidence intervals*

## C Considerations for the choice of $W$ and $Z$ variables

We suppose that variables $X$ are partitioned into baseline variables $W$ and conditional covariates $Z$. We suggest selecting $Z$ to be the variables that may act as mediators of the environment shift and/or variables whose associations with $Y$ are likely to be modified by the environment shift (i.e. effect modification). This selection can be chosen based on a high-level causal graph, where $W$ are variables known to be upstream of $Z$. For instance, if $Z$ are treatment variables and $W$ are baseline variables, one can interpret a covariate shifts as a change in the treatment policy and an outcome shift as a change in the treatment effect across the two environments.

In the absence of any prior knowledge, another option is to choose $W$ as the variables for which one would like the expected loss given $W$ to be invariant across the two environments; this can be useful to promote fairness of ML algorithms across environments. When this invariance does not hold, the proposed framework explains how variables $Z$ contribute to these differences, which can inform efforts to eliminate performance gaps. This last option is similar to how variables are typically chosen in disparity analyses [21]. For instance, to understand why income differs between males and females controlling for age, one would set domain to $D = $ gender, $W = $ age, and $Z$ as variables that may explain this disparity (e.g. marital status, employment status).

In general, including more variables in $W$ and $Z$ to explain the performance difference is preferable. Nevertheless, there are tradeoffs. For instance, including more variables in $W$ leads to higher variance of $\Delta_{\cdot 10}(W)$, so it allows one to better distinguish the relative importance of variables in $Z$ for explaining its variability. On the other hand, when more variables are assigned to $W$, the performance gap with respect to $W$ ($\Delta_{\cdot 10}(W)$) is a more complex function. Thus we may have more uncertainty in our estimate of $\Delta_{\cdot 10}(W)$, which may lead to wider confidence intervals for the variable importance values.

## D Causal interpretation of aggregate and detailed decompositions

Partial distribution shifts that we define in the framework can be equivalently described as stochastic (or shift) interventions in a structural causal model (SCM) respecting a causal directed acyclic graph (DAG) [8]. To represent an intervention on variable $X$, we use regime indicator $\sigma_X$ which means that the conditional probability distribution for $X$ in the SCM has been updated to a new one [9].

Suppose that the source and target data $(W, Z, Y)$ are generated by SCMs respecting the same DAG $\mathcal{G}$ in Figure 4 with no unmeasured confounders. Intervening on a variable in source SCM sets its

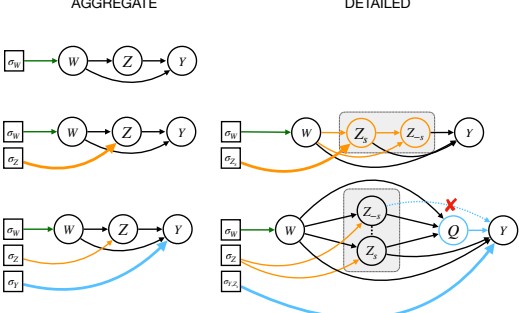

Figure 4: Decomposition framework for explaining the performance gap from source to target domain, visualized through causal directed acyclic graphs. Aggregate decompositions describe the incremental effect of stochastic interventions on each aggregate variable's distribution at the source with that in the target, indicated by regime indicators $\sigma_{\mathtt{W}}, \sigma_{\mathtt{Z}}$, and $\sigma_{\mathtt{Y}}$. Detailed decompositions quantify how well candidate partial distribution shifts explain the variability of performance gaps across strata. The candidate partial shifts considered in this work are shown in the DAGs on the right. An $s$-partial conditional covariate shift is a stochastic intervention on variable subset $Z_s$. An $s$-partial conditional outcome shift is a stochastic intervention on variable $Y$, in which the conditional outcome distribution fine-tunes the risk in the source domain (indicated by the additional node $Q = p_0(Y = 1|W, Z)$) with respect to $Z_s$.

conditional distribution to the corresponding one in the target SCM. Under the assumption of no unmeasured confounders, we have

$$p_0(V; \sigma_X) = \prod_{V_i \in V \setminus X} p_0(V_i|pa_{\mathcal{G}}(V_i); \sigma_X) \prod_{V_i \in X} p_1(V_i|pa_{\mathcal{G}}(V_i)) \tag{10}$$

for any variable set $V$ where $pa_{\mathcal{G}}$ denotes parents in $\mathcal{G}$. Therefore, the target data is obtained by intervening on all the variables $p_0(W, Z, Y; \sigma_{\mathtt{W}}, \sigma_{\mathtt{Z}}, \sigma_{\mathtt{Y}}) = p_1(W, Z, Y)$. Expectation $\mathbb{E}$ is taken over the source distribution or its shifted version based on the specified regime indicator.

Aggregate decompositions can then be written as causal effect of intervening on $W, Z, Y$ incrementally as follows

$$\Lambda_{\mathtt{W}} = \mathbb{E}[\ell|\sigma_{\mathtt{W}}] - \mathbb{E}[\ell]$$
$$\Lambda_{\mathtt{Z}} = \mathbb{E}[\ell|\sigma_{\mathtt{W}}, \sigma_{\mathtt{Z}}] - \mathbb{E}[\ell|\sigma_{\mathtt{W}}]$$
$$\Lambda_{\mathtt{Y}} = \mathbb{E}[\ell|\sigma_{\mathtt{W}}, \sigma_{\mathtt{Z}}, \sigma_{\mathtt{Y}}] - \mathbb{E}[\ell|\sigma_{\mathtt{W}}, \sigma_{\mathtt{Z}}]$$

Due to the factorization (10), the above are equivalent to aggregate decompositions presented in Section 2.

For detailed decomposition of the conditional covariate shift, assume the causal DAG in Figure 4 with $Z_s \to Z_{-s}$ for variable subset $s$. The $s$-partial conditional covariate shift is represented by $p_0(V; \sigma_{Z_s})$ and, under the factorization (10), is equivalent to the one considered in Section 2.2. Thus, the performance difference $\Delta_{\cdot s0}(W)$ can be equivalently written as $\mathbb{E}[\ell|W; \sigma_{Z_s}] - \mathbb{E}[\ell|W]$ and $\Delta_{\cdot 10}(W)$ as $\mathbb{E}[\ell|W; \sigma_{\mathtt{Z}}] - \mathbb{E}[\ell|W]$. The value function is then

$$v_{\mathtt{Z}}(s) := 1 - \frac{\mathbb{E}\left[(\Delta_{\cdot s0}(W) - \Delta_{\cdot 10}(W))^2; \sigma_{\mathtt{W}}\right]}{\mathbb{E}\left[\Delta_{\cdot 10}^2(W); \sigma_{\mathtt{W}}\right]}.$$

The $s$-partial conditional outcome shift corresponds to a stochastic intervention that changes the distribution of $Y$ to (5). Denoting it as $p_0(V; \sigma_{Y, Z_s})$, the performance difference $\Delta_{\cdot \cdot s}(W, Z)$ can be equivalently written as $\mathbb{E}[\ell|W, Z; \sigma_{Y, Z_s}] - \mathbb{E}[\ell|W, Z]$ and $\Delta_{\cdot \cdot 1}(W, Z)$ as $\mathbb{E}[\ell|W, Z; \sigma_{\mathtt{Y}}] - \mathbb{E}[\ell|W, Z]$. The value function (4) is then

$$v_{\mathtt{Y}}(s) := 1 - \frac{\mathbb{E}\left[(\Delta_{\cdot \cdot s}(W, Z) - \Delta_{\cdot \cdot 1}(W, Z))^2; \sigma_{\mathtt{W}}, \sigma_{\mathtt{Z}}\right]}{\mathbb{E}\left[\Delta_{\cdot \cdot 1}^2(W, Z); \sigma_{\mathtt{W}}, \sigma_{\mathtt{Z}}\right]}. \tag{11}$$

# E  Proper scoring rules for partial distribution shift

Prior works have considered scoring rules for partial distribution shifts in terms of their change in the average loss [48]. For instance, for conditional covariate shifts, prior works have considered scoring rules of the form

$$\mathbb{E}_{1s0}[\ell] - \mathbb{E}_{100}[\ell] = \int \ell(Y, f(Z, W)) \left( p_s(z|w) - p_0(z|w) \right) p_1(w) dz dw. \tag{12}$$

However, in the absence of a detailed causal graph, we show that this can lead to unintuitive attributions. Consider the following counterexample. We drop $W$ from the data example for simplicity of exposition.

*Example* E.1. Consider the following data-generating process with random variables $Z_1$ and $Z_2$, a real-valued outcome $Y$, and loss function as squared error $\ell = (f(Z_1, Z_2) - Y)^2$:

$$Z_1 \sim N(D+1, 1) \tag{13}$$
$$Z_2 = |Z_1| \tag{14}$$
$$\epsilon \sim N(0, 1) \tag{15}$$
$$Y = Z_1 + Z_2 + \epsilon \mathbb{1}\{Z_1 \leq 0\} \tag{16}$$

where $D = 0$ and $D = 1$ correspond to the source and target domains, respectively. Suppose the ML model is the optimal model for minimizing the expected loss in the source domain, i.e. $f(Z_1, Z_2) = Z_1 + Z_2$. Consider the following candidate partial distribution shifts for explaining the performance change, where Option 1 hypothesizes $Z_1$ causes $Z_2$ and Option 2 hypothesizes $Z_2$ causes $Z_1$.

**(Option 1)** For $s = \{1\}, p_s(Z) = p_1(Z_1)p_0(Z_2|Z_1)$

**(Option 2)** For $s = \{2\}, p_s(Z) = p_1(Z_2)p_0(Z_1|Z_2)$

Per the given data-generating process, the partial shift in Option 1 exactly corresponds to the true (aggregate) dataset shift; thus we would expect it to have a higher value than Option 2. Nevertheless, the MSE resulting from Option 2's shift is 0.5 (the marginal distribution of $Z_1$ under Option 2 is symmetric around 0). In contrast, the MSE resulting from Option 1's shift is the probability that a standard normal variable is less than -1, which is 0.159.

In Example E.1, scoring rule (12) assigns a higher value to a partial shift that contradicts the true causal graph than even the true aggregate shift, because the rule *assumes* the hypothesized dataset shift is true. Consequently, (12) is not a *proper* scoring rule. In contrast, the proposed value functions described in Section 2.1 are.

More formally, we extend the traditional definition for a proper scoring rule [18] to the context of explaining performance changes as follows. Let a scoring function $\chi : O_0 \times O_1 \times Q$ be defined as a function of source observation $O_0 = (W_0, Z_0, Y_0)$, target observation $O_1 = (W_1, Z_1, Y_1)$, and candidate shift probability model $Q$. A scoring rule $\chi$ is proper with respect to the set of distribution shift models $\mathcal{F}$ if the following holds: for any true model of the post-shift data distribution $p^*(W, Z, Y) \in \mathcal{F}$, the expectation of the scoring function $\chi(O_0, O_1, p^*)$ with respect to $O_0 \sim p_0$ and $O_1 \sim p^*$ is always no smaller than the expectation of $\chi(O_0, O_1, p)$ for any $p \in \mathcal{F}$, i.e.

$$\mathbb{E}_{p_0, p^*}[\chi(O_0, O_1, p^*)] \geq \mathbb{E}_{p_0, p^*}[\chi(O_0, O_1, p)] \quad \forall p \in \mathcal{F} \tag{17}$$

Moreover, we say that $\chi$ is strictly proper if (17) holds with equality iff $p = p^*$.

Let $\mathcal{F}_{\mathsf{Z}}$ be the set of candidate $s$-partial conditional covariate shifts $p_{\cdot s0}$ and $\mathcal{F}_{\mathsf{Y}}$ be the set of candidate $s$-partial conditional outcome shifts $p_{\cdot \cdot s}$, i.e.

$$\mathcal{F}_{\mathsf{Z}} = \{p_{\cdot s0}(y, z|w) = p_0(y|z, w)p_s(z|w) : s \subseteq \{1, \cdots, m_2\}\}$$
$$\mathcal{F}_{\mathsf{Y}} = \{p_s(y|z, w) : s \subseteq \{1, \cdots, m_2\}\}.$$

Then, $v_{\mathsf{Z}}$ defined in (3) is a proper scoring rule with respect to $\mathcal{F}_{\mathsf{Z}}$, as $v_{\mathsf{Z}}$ attains its largest value when $\Delta_{\cdot s0} \equiv \Delta_{\cdot 10}$, which holds when $p_{\cdot s0}$ matches the true covariate-only distribution shift model $p_{\cdot 10}$. Similarly, $v_{\mathsf{Y}}$ defined in (4) is a proper scoring rule with respect to $\mathcal{F}_{\mathsf{Y}}$. Moreover, these are *strictly* proper scoring rules as long as the conditional expectation of the losses of candidate distribution shifts are not adversarially aligned. That is, under the assumptions that

$$\mathbb{E}_{\cdot s0}[\ell|W] \equiv \mathbb{E}_{\cdot 10}[\ell|W] \iff p_{\cdot s0} \equiv p_{\cdot 10} \quad \forall p_{\cdot s0} \in \mathcal{F}_{\mathsf{Z}} \tag{A.1}$$
$$\mathbb{E}_{\cdot \cdot s}[\ell|W, Z] \equiv \mathbb{E}_{\cdot \cdot 1}[\ell|W, Z] \iff p_{\cdot \cdot s} \equiv p_{\cdot \cdot 1} \quad \forall p_{\cdot \cdot s} \in \mathcal{F}_{\mathsf{Y}} \tag{A.2}$$

$v_{\mathtt{Z}}$ and $v_{\mathtt{Y}}$ are strictly proper scoring rules with respect to $\mathcal{F}_{\mathtt{Z}}$ and $\mathcal{F}_{\mathtt{Y}}$, respectively. As the candidate distribution shifts considered in this work are functionals of the observed distribution shift and observed distribution shifts are unlikely to be adversarial in many real-world situations, Assumptions (A.1) and (A.2) are likely reasonable in practice.

# F Estimation and Inference

Let $\{(W_i^{(d)}, Z_i^{(d)}, Y_i^{(d)}) : i = 1, \cdots, n_d\}$ denote $n_d$ independent and identically distributed (IID) observations from the source and target domains $d = 0$ and $1$, respectively. Let a fixed fraction of the data be partitioned towards "training" (Tr) and the remaining to "evaluation" (Ev); let $n_{\mathtt{Ev}}$ be the number of observations in the evaluation partition. Let $\mathbb{P}_d$ denote the expectation with respect to domain $d$ and $\mathbb{P}_{d,n}$ denote the empirical average over observations in partition Ev from domain $d = \{0, 1\}$.

## F.1 Aggregate decomposition

The aggregate decomposition terms can be formulated as an average treatment effect, a well-studied estimand in causal inference, where domain $D$ corresponds to treatment. As such, one can use augmented inverse probability weighting (AIPW) to define debiased ML estimators of the aggregate decomposition terms (e.g. Kang and Schafer [24]). We review estimation and inference for these terms below.

**Estimation.** Using the training data, estimate outcome models $\mu_{\cdot 00}(W) = \mathbb{E}_{\cdot 00}[\ell | W = w]$ and $\mu_{\cdot \cdot 0}(W, Z) = \mathbb{E}[\ell | W, Z]$ and density ratio models $\pi_{100}(W) = p_1(W)/p_0(W)$ and $\pi_{110}(W, Z) = p_1(W, Z)/p_0(W, Z)$. The debiased ML estimators for $\Lambda_{\mathtt{W}}, \Lambda_{\mathtt{Z}}, \Lambda_{\mathtt{Y}}$ are

$$\hat{\Lambda}_{\mathtt{W}} = \mathbb{P}_{0,n}\left(\ell - \hat{\mu}_{\cdot 00}(W)\right)\hat{\pi}_{100}(W) + \mathbb{P}_{1,n}\hat{\mu}_{\cdot 00}(W) - \mathbb{P}_{0,n}\ell$$

$$\hat{\Lambda}_{\mathtt{Z}} = \mathbb{P}_{0,n}\left(\ell - \hat{\mu}_{\cdot \cdot 0}(W, Z)\right)\hat{\pi}_{110}(W, Z) + \mathbb{P}_{1,n}\hat{\mu}_{\cdot \cdot 0}(W, Z)$$
$$\quad - \mathbb{P}_{0,n}\left(\ell - \hat{\mu}_{\cdot 00}(W)\right)\hat{\pi}_{100}(W) - \mathbb{P}_{1,n}\hat{\mu}_{\cdot 00}(W)$$

$$\hat{\Lambda}_{\mathtt{Y}} = \mathbb{P}_{1,n}\ell - \mathbb{P}_{0,n}\left(\ell - \hat{\mu}_{\cdot \cdot 0}(W, Z)\right)\hat{\pi}(W, Z) - \mathbb{P}_{1,n}\hat{\mu}_0(W, Z)$$

**Inference.** Assuming the estimators for the outcome and density ratio models converge at a fast enough rate, the AIPW estimators for the aggregate decomposition terms are asymptotically linear and, thus, facilitate the construction of CIs.

**Theorem F.1.** *Suppose $\pi_{100}$ and $\pi_{110}$ are bounded; estimators $\hat{\mu}_{\cdot 00}$, $\hat{\pi}_{\cdot \cdot 0}$, $\hat{\pi}_{100}$, and $\hat{\pi}_{110}$ are consistent; and*

$$\mathbb{P}_0\left(\hat{\mu}_{\cdot 00} - \mu_{\cdot 00}\right)\left(\hat{\pi}_{100} - \pi_{100}\right) = o_p(n^{-1/2})$$
$$\mathbb{P}_0\left(\hat{\mu}_{\cdot \cdot 0} - \mu_{\cdot \cdot 0}\right)\left(\hat{\pi}_{110} - \pi_{110}\right) = o_p(n^{-1/2}).$$

*Then $\hat{\Lambda}_{\mathtt{W}}, \hat{\Lambda}_{\mathtt{Z}},$ and $\hat{\Lambda}_{\mathtt{Y}}$ are asymptotically linear estimators of their respective estimands.*

## F.2 Value of $s$-partial conditional covariate shifts

**Estimation.** Using the training partition, estimate the density ratio $\pi_{1s0}(z_s, w) = p_1(z_s, w)/p_0(z_s, w)$ and the outcome models

$$\mu_{\cdot 0_{-s}0}(z_s, w) = \mathbb{E}_{\cdot 00}[\ell | z_s, w] = \int\int \ell p_0(y|w, z)p_0(z_{-s}|z_s, w)dydz_{-s} \tag{18}$$

$$\mu_{\cdot 10}(w) = \mathbb{E}_{\cdot 10}[\ell | w] \tag{19}$$

$$\mu_{\cdot s0}(w) = \mathbb{E}_{\cdot s0}[\ell | w] = \int\int\int \ell p_0(y|w, z)p_0(z_{-s}|z_s, w)p_1(z_s|w)dydz_{-s}dz_s, \tag{20}$$

in addition to the other nuisance models previously mentioned. We propose the estimator $\hat{v}_{\mathtt{Z}}(s) = \hat{v}_{\mathtt{Z}}^{\mathtt{num}}(s)/\hat{v}_{\mathtt{Z}}^{\mathtt{den}}$, where

$$
\begin{aligned}
\hat{v}_{\mathtt{Z}}^{\mathtt{num}}(s) :=& \mathbb{P}_{1,n}(\hat{\mu}_{\cdot s0}(W) - \hat{\mu}_{\cdot 10}(W))^2 \\
&+ 2\mathbb{P}_{0,n}(\hat{\mu}_{\cdot s0}(W) - \hat{\mu}_{\cdot 10}(W))(\ell - \hat{\mu}_{\cdot 0_{-s}0}(W, Z_s))\hat{\pi}_{1s0}(W, Z_s) \\
&- 2\mathbb{P}_{0,n}(\hat{\mu}_{\cdot s0}(W) - \hat{\mu}_{\cdot 10}(W))(\ell - \hat{\mu}_{\cdot\cdot 0}(W, Z))\hat{\pi}_{110}(W, Z) \\
&+ 2\mathbb{P}_{1,n}(\hat{\mu}_{\cdot s0}(W) - \hat{\mu}_{\cdot 10}(W))(\hat{\mu}_{\cdot 0_{-s}0}(W, Z_s) - \hat{\mu}_{\cdot s0}(W)) \\
&- 2\mathbb{P}_{1,n}(\hat{\mu}_{\cdot s0}(W) - \hat{\mu}_{\cdot 10}(W))(\hat{\mu}_{\cdot\cdot 0}(W, Z) - \hat{\mu}_{\cdot 10}(W))
\end{aligned}
\tag{21}
$$

and $\hat{v}_{\mathtt{Z}}^{\mathtt{den}} := \hat{v}_{\mathtt{Z}}^{\mathtt{num}}(\emptyset)$.

**Inference.** The estimator is asymptotically normal as long as the outcome and density ratio models are estimated at a fast enough rate defined formally as follows.

**Condition F.2.** *For variable subset $s$, suppose the following holds*

- $\mathbb{P}_0(\mu_{001}(W) - \mu_{01}(W))^2$ *is bounded*

- $\mathbb{P}_0(\mu_{\cdot 0_{-s}0}(Z_s, W) - \hat{\mu}_{\cdot 0_{-s}0}(Z_s, W))(\hat{\pi}_{1s0}(Z_s, W) - \pi_{1s0}(Z_s, W)) = o_p(n^{-1/2})$

- $\mathbb{P}_0(\mu_{\cdot\cdot 0}(W, Z) - \hat{\mu}_{\cdot\cdot 0}(W, Z))(\hat{\pi}_{110}(W, Z) - \pi_{110}(W, Z)) = o_p(n^{-1/2})$

- $\mathbb{P}_1(\hat{\mu}_{\cdot s0}(W) - \mu_{\cdot s0}(W))^2 = o_p(n^{-1/2})$

- $\mathbb{P}_1(\hat{\mu}_{\cdot 10}(W) - \mu_{\cdot 10}(W))^2 = o_p(n^{-1/2})$

- *(Positivity)* $p_0(z_s, w) > 0$ *and* $p_0(w, z) > 0$ *almost everywhere, such that the density ratios* $\pi_{1s0}(w, z_s)$ *and* $\pi_{110}(w, z)$ *are well-defined and between* $(0, 1)$.

**Theorem F.3.** *For variable subset $s$, suppose $v_Z^{den}(s) > 0$ and Condition F.2 hold. Then the estimator $\hat{v}_Z(s)$ is asymptotically normal.*

## G  Proofs

**Notation.** For all proofs, we will write $\mathbb{P}$ to mean expectation on the evaluation partition (and likewise for the empirical version) for notational simplicity.

**Overview of derivation strategy.** We first present the general strategy for proving asymptotic normality of the estimators for the decompositions. Details on nonparametric debiased estimation can be found in texts such as Tsiatis [45] and Kennedy [25].

Let $v(\mathbb{P})$ be a pathwise differentiable quantity that is a function of the true regular (differentiable in quadratic mean) probability distribution $\mathbb{P}$ over random variable $O$. For instance, $v$ in the case of mean is defined as $v(\mathbb{P}) := \mathbb{E}_{o \sim \mathbb{P}(O)}[o]$. Let $\hat{\mathbb{P}}$ denote an arbitrary regular estimator of $\mathbb{P}$, such as the maximum likelihood estimator. The plug-in estimator is then defined as $v(\hat{\mathbb{P}})$.

The von-Mises expansion of the functional $v$ (which linearizes $v$ in analogy to the first-order Taylor expansion), given it is pathwise differentiable, gives

$$
v(\hat{\mathbb{P}}) - v(\mathbb{P}) = -\mathbb{P}\,\psi(o; \hat{\mathbb{P}}) + R(\hat{\mathbb{P}}, \mathbb{P}).
\tag{22}
$$

Here, the function $\psi$ is called an influence function (or a functional gradient of $v$ at $\hat{\mathbb{P}}$). $R(\hat{\mathbb{P}}, \mathbb{P})$ is a second-order remainder term. The one-step corrected estimators we consider have the form of $v(\hat{\mathbb{P}}) + \mathbb{P}_n\psi(o; \hat{\mathbb{P}})$ where $\mathbb{P}_n$ denotes a sample average. Following the expansion above, the one-step corrected estimator can be analyzed as follows,

$$
\begin{aligned}
&\left(v(\hat{\mathbb{P}}) + \mathbb{P}_n\psi(o; \hat{\mathbb{P}})\right) - v(\mathbb{P}) \\
&= (\mathbb{P}_n - \mathbb{P})\psi(o; \hat{\mathbb{P}}) + R(\hat{\mathbb{P}}, \mathbb{P}) \\
&= (\mathbb{P}_n - \mathbb{P})\psi(o; \mathbb{P}) + (\mathbb{P}_n - \mathbb{P})(\psi(o; \hat{\mathbb{P}}) - \psi(o; \mathbb{P})) + R(\hat{\mathbb{P}}, \mathbb{P})
\end{aligned}
$$

Our goal will be to analyze each of the three terms and to show that they are asymptotically negligible at $\sqrt{n}$-rate, such that the one-step corrected estimator satisfies

$$\left(v(\hat{\mathbb{P}}) + \mathbb{P}_n\psi(o;\hat{\mathbb{P}})\right) - v(\mathbb{P}) = \mathbb{P}_n\psi(o;\mathbb{P}) + o_p(n^{-1/2}),$$

where we used the property of influence functions that they have zero mean. Thus the one-step corrected estimator is asymptotically normal with mean $v(\mathbb{P})$ and variance $var(\psi(o;\mathbb{P}))/n$, which allows for the construction of CIs. In the following proofs, we present the influence functions without derivations; see Kennedy [25] and Hines et al. [20] for strategies for deriving influence functions.

### G.1 Aggregate decompositions

Let the nuisance parameters in the one-step estimators $\Lambda_W, \Lambda_Z, \Lambda_Y$ be denoted by $\eta_W = (\mu_{\cdot 00}, \pi_{100}), \eta_Z = (\mu_{\cdot\cdot 0}, \mu_{\cdot 00}, \pi_{110}), \eta_Y = (\mu_{\cdot\cdot 0}, \pi_{110})$ respectively. Denote the estimated nuisances by $\hat{\eta}_W, \hat{\eta}_Z, \hat{\eta}_Y$. The canonical gradients for the three estimands are

$$\psi_W(d, w, z, y; \eta_W) = [(\ell(w, z, y) - \mu_{\cdot 00}(w))\,\pi_{100}(w) - \ell(w, z, y)]\frac{\mathbb{1}\{d = 0\}}{p(d = 0)}$$

$$+ \mu_{\cdot 00}(w)\frac{\mathbb{1}\{d = 1\}}{p(d = 1)} - \Lambda_W \tag{23}$$

$$\psi_Z(d, w, z, y; \eta_Z) = [(\ell(w, z, y) - \mu_{\cdot\cdot 0}(w, z))\,\pi_{110}(w, z)]\frac{\mathbb{1}\{d = 0\}}{p(d = 0)} + \mu_{\cdot\cdot 0}(w, z)\frac{\mathbb{1}\{d = 1\}}{p(d = 1)}$$

$$- [(\ell(w, z, y) - \mu_{\cdot 00}(w))\,\pi_{100}(w)]\frac{\mathbb{1}\{d = 0\}}{p(d = 0)} + \mu_{\cdot 00}(w)\frac{\mathbb{1}\{d = 1\}}{p(d = 1)} - \Lambda_Z \tag{24}$$

$$\psi_Y(d, w, z, y; \eta_Y) = (\ell(w, z, y) - \mu_{\cdot\cdot 0}(w, z))\frac{\mathbb{1}\{d = 1\}}{p(d = 1)}$$

$$- [(\ell(w, z, y) - \mu_{\cdot\cdot 0}(w, z))\,\pi_{110}(w, z)]\frac{\mathbb{1}\{d = 0\}}{p(d = 0)} - \Lambda_Y. \tag{25}$$

**Theorem G.1** (Theorem F.1). *Under conditions outlined in Theorem F.1, the one-step corrected estimators for the aggregate decomposition terms, baseline, conditional covariate, and conditional outcome $\hat{\Lambda}_W$, $\hat{\Lambda}_Z$, and $\hat{\Lambda}_Y$, are asymptotically linear, i.e.*

$$\hat{\Lambda}_N - \Lambda_N = \mathbb{P}_n\psi_N + o_p(n^{-1/2}) \quad \forall N \in \{W, Z, Y\}. \tag{26}$$

*Proof.* The estimands $\Lambda_W, \Lambda_Z, \Lambda_Y$ have similarities to the standard average treatment effect (ATE) in the causal inference literature (see [25, Example 2]. Hence, the estimators and their asymptotic properties directly follow. For treatment $T$, outcome $O$, and confounders $C$, the mean outcome under $T = 1$ among the population with $T = 0$ is identified as

$$\phi = \int op(o|c, t = 1)p(c|t = 0)dodc \tag{27}$$

and its one-step corrected estimator can be derived from the canonical gradient of $\phi$, which takes the following form after plugging in the estimates of the nuisance models:

$$\hat{\phi}_n = \mathbb{P}_n\left\{\frac{\mathbb{1}\{T = 1\}}{\Pr(T = 1)}\hat{\pi}(c)\,(O - \hat{\mu}_1(c)) + \frac{\mathbb{1}\{T = 0\}}{\Pr(T = 0)}\hat{\mu}_1(c)\right\}$$

satisfies

$$\hat{\phi}_n - \phi = \mathbb{P}_n\left\{\frac{\mathbb{1}\{T = 1\}}{\Pr(T = 1)}\pi(c)\,(O - \mu_1(c)) + \frac{\mathbb{1}\{T = 0\}}{\Pr(T = 0)}\mu_1(c) - \phi\right\} + o_p(n^{-1/2})$$

where $\mu_1(c) = \mathbb{E}[o|c, t = 1]$ and $\pi(c) = p(c|t = 0)/p(c|t = 1)$ as long as the following conditions hold:

- $p(c|t = 1) > 0$ almost everywhere such that the density ratios $\pi(c)$ are well-defined and bounded,

- $\mathbb{P}_1(\hat{\mu}_1 - \mu_1)(\hat{\pi} - \pi) = o_p(n^{-1/2})$.

We establish the estimators and their influence functions by showing that they can all be viewed as mean outcomes of the form (27).

**Baseline term** $\Lambda_{\mathtt{W}}$.  The first term $\mathbb{E}_{100}\left[\ell(w,z,y)\right]$ is a mean outcome with respect to $p(\ell(w,z,y)|w,d=0)p(w|d=1)$, which is the same as that in (27) but with $\ell(w,z,y)$ as the outcome, $w$ as the confounder, and $d$ as the (flipped) treatment. The second term $\mathbb{E}_{000}\left[\ell(w,z,y)\right]$ is a simple average over $D=0$ population whose influence function is the $\ell(w,z,y)$ itself.

**Conditional covariate term** $\Lambda_{\mathtt{Z}}$. First term $\mathbb{E}_{110}\left[\ell(w,z,y)\right]$ is the mean outcome with respect to $p(\ell(w,z,y)|w,z,d=0)p(w,z|d=1)$, where the chief difference is $(w,z)$ is the confounder. Second term $\mathbb{E}_{100}\left[\ell(w,z,y)\right]$ is also a mean outcome, as discussed above.

**Conditional outcome term** $\Lambda_{\mathtt{Y}}$. First term $\mathbb{E}_{111}\left[\ell(w,z,y)\right]$ is a simple average over the $D=1$ population. $\square$

### G.2 Value of $s$-partial conditional covariate shifts

Let nuisance parameters in the one-step estimator $v_{\mathtt{Z},s}^{\mathrm{num}}$ be denoted $\eta_{\mathtt{Z},s}^{\mathrm{num}} = \left(\mu_{\cdot s0}, \mu_{\cdot 10}, \mu_{\cdot 0_{-s}0}, \mu_{001}, \mu_{\cdot\cdot 0}, \pi_{1s0}, \pi_{110}\right)$ and the set of estimated nuisances by $\hat{\eta}_{\mathtt{Z},s}^{\mathrm{num}}$.  The canonical gradient of $v_{\mathtt{Z}}^{\mathrm{num}}(s)$ is

$$
\psi_{\mathtt{Z},s}^{\mathrm{num}}(D,W,Z,Y;\eta_{\mathtt{Z},s}^{\mathrm{num}}) = (\mu_{\cdot s0}(W) - \mu_{\cdot 10}(W))^2 \frac{\mathbb{1}\{D=1\}}{\Pr(D=1)}
$$

$$
+ 2(\mu_{\cdot s0}(W) - \mu_{\cdot 10}(W))(\ell - \mu_{\cdot 0_{-s}0}(W,Z_s))\pi_{1s0}(W,Z_s)\frac{\mathbb{1}\{D=0\}}{\Pr(D=0)}
$$

$$
- 2(\mu_{\cdot s0}(W) - \mu_{\cdot 10}(W))(\ell - \mu_{\cdot\cdot 0}(W,Z))\pi_{110}(W,Z)\frac{\mathbb{1}\{D=0\}}{\Pr(D=0)}
$$

$$
+ 2(\mu_{\cdot s0}(W) - \mu_{\cdot 10}(W))(\mu_{\cdot 0_{-s}0}(W,Z_s) - \mu_{\cdot s0}(W))\frac{\mathbb{1}\{D=1\}}{\Pr(D=1)}
$$

$$
- 2(\mu_{\cdot s0}(W) - \mu_{\cdot 10}(W))(\mu_{\cdot\cdot 0}(W,Z) - \mu_{\cdot 10}(W))\frac{\mathbb{1}\{D=1\}}{\Pr(D=1)}
$$

$$
- v_{\mathtt{Z}}^{\mathrm{num}}(s). \tag{28}
$$

**Lemma G.2.** *Under Condition F.2, $\hat{v}_{\mathtt{Z}}^{num}(s)$ satisfies*

$$
\hat{v}_{\mathtt{Z}}^{num}(s) - v_{\mathtt{Z}}^{num}(s) = \mathbb{P}_n \psi_{\mathtt{Z},s}^{num}(D,W,Z,Y;\eta_{\mathtt{Z},s}^{num}) + o_p(n^{-1/2}) \tag{29}
$$

$$
\hat{v}_{\mathtt{Z}}^{den} - v_{\mathtt{Z}}^{den} = \mathbb{P}_n \psi_{\mathtt{Z},\emptyset}^{num}(D,W,Z,Y;\eta_{\mathtt{Z},s}^{num}) + o_p(n^{-1/2}) \tag{30}
$$

*Proof.* Consider the following decomposition

$$
\hat{v}_{\mathtt{Z}}^{\mathrm{num}}(s) - v_{\mathtt{Z}}^{\mathrm{num}}(s)
$$
$$
= (\mathbb{P}_n - \mathbb{P})\psi_{\mathtt{Z}}^{\mathrm{num}}(D,W,Z,Y;\eta_{\mathtt{Z},s}^{\mathrm{num}}) \tag{31}
$$
$$
+ (\mathbb{P}_n - \mathbb{P})(\psi_{\mathtt{Z}}^{\mathrm{num}}(D,W,Z,Y;\hat{\eta}_{\mathtt{Z},s}^{\mathrm{num}}) - \psi_{\mathtt{Z}}^{\mathrm{num}}(W,Z,Y;\eta_{\mathtt{Z},s}^{\mathrm{num}})) \tag{32}
$$
$$
+ \mathbb{P}(\psi_{\mathtt{Z}}^{\mathrm{num}}(D,W,Z,Y;\hat{\eta}_{\mathtt{Z},s}^{\mathrm{num}}) - \psi_{\mathtt{Z}}^{\mathrm{num}}(D,W,Z,Y;\eta_{\mathtt{Z},s}^{\mathrm{num}})) \tag{33}
$$

We note that (32) converges to a normal distribution per CLT assuming the variance of $\psi_{\mathtt{Z},s}^{\mathrm{num}}$ is finite. The empirical process term (32) is asymptotically negligible, as the nuisance parameters $\eta_{\mathtt{Z},s}^{\mathrm{num}}$ are estimated using a separate training data split from the evaluation data and [25, Lemma 1] states that

$$
(\mathbb{P}_n - \mathbb{P})(\psi_{\mathtt{Z},s}^{\mathrm{num}}(D,W,Z,Y;\hat{\eta}_{\mathtt{Z},s}^{\mathrm{num}}) - \psi_{\mathtt{Z},s}^{\mathrm{num}}(D,W,Z,Y;\eta_{\mathtt{Z},s}^{\mathrm{num}}) = o_p(n^{-1/2})
$$

as long as estimators for all nuisance parameters are consistent. We now establish that the remainder term (33) is also asymptotically negligible. Integrating with respect to $Y$, we have that

$$(33) = \mathbb{P}\, 2(\hat{\mu}_{\cdot s0}(W) - \hat{\mu}_{\cdot 10}(W)) \times \Big( (\mu_{\cdot 0_{-s}0}(Z_s, W) - \hat{\mu}_{\cdot 0_{-s}0}(Z_s, W))\hat{\pi}_{1s0}(W, Z_s)\frac{\mathbb{1}\{D=0\}}{p(D=0)}$$

$$+ (\hat{\mu}_{\cdot 0_{-s}0}(Z_s, W) - \hat{\mu}_{\cdot s0}(W))\frac{\mathbb{1}\{D=1\}}{p(D=1)}$$

$$- (\mu_{\cdot\cdot 0}(W, Z) - \hat{\mu}_{\cdot\cdot 0}(W, Z))\hat{\pi}_{110}(W, Z)\frac{\mathbb{1}\{D=0\}}{p(D=0)}$$

$$- (\hat{\mu}_{\cdot\cdot 0}(W, Z) - \hat{\mu}_{\cdot 10}(W))\frac{\mathbb{1}\{D=1\}}{p(D=1)}\Big)$$

$$\tag{34}$$

$$+ \mathbb{P}((\hat{\mu}_{\cdot s0}(W) - \hat{\mu}_{\cdot 10}(W))^2 - (\mu_{\cdot s0}(W) - \mu_{\cdot 10}(W))^2)\frac{\mathbb{1}\{D=1\}}{p(D=1)} \tag{35}$$

$$= \mathbb{P}\, 2(\hat{\mu}_{\cdot s0}(W) - \hat{\mu}_{\cdot 10}(W)) \times \Big( (\mu_{\cdot 0_{-s}0}(Z_s, W) - \hat{\mu}_{\cdot 0_{-s}0}(Z_s, W))(\hat{\pi}_{1s0}(W, Z_s) - \pi_{1s0}(W, Z_s))\frac{\mathbb{1}\{D=0\}}{p(D=0)}$$

$$+ (\mu_{\cdot 0_{-s}0}(Z_s, W) - \hat{\mu}_{\cdot 0_{-s}0}(Z_s, W))\pi_{1s0}(W, Z_s)\frac{\mathbb{1}\{D=0\}}{p(D=0)}$$

$$+ (\hat{\mu}_{\cdot 0_{-s}0}(Z_s, W) - \hat{\mu}_{\cdot s0}(W))\frac{\mathbb{1}\{D=1\}}{p(D=1)}$$

$$- (\mu_{\cdot\cdot 0}(W, Z) - \hat{\mu}_{\cdot\cdot 0}(W, Z))(\hat{\pi}_{110}(W, Z) - \pi_{110}(W, Z))\frac{\mathbb{1}\{D=0\}}{p(D=0)}$$

$$- (\mu_{\cdot\cdot 0}(W, Z) - \hat{\mu}_{\cdot\cdot 0}(W, Z))\pi_{110}(W, Z)\frac{\mathbb{1}\{D=0\}}{p(D=0)}$$

$$- (\hat{\mu}_{\cdot\cdot 0}(W, Z) - \hat{\mu}_{\cdot 10}(W))\frac{\mathbb{1}\{D=1\}}{p(D=1)}\Big)$$

$$\tag{36}$$

$$+ \mathbb{P}((\hat{\mu}_{\cdot s0}(W) - \hat{\mu}_{\cdot 10}(W))^2 - (\mu_{\cdot s0}(W) - \mu_{\cdot 10}(W))^2)\frac{\mathbb{1}\{D=1\}}{p(D=1)} \tag{37}$$

From convergence conditions in Condition F.2, this simplifies to

$$(33) = \mathbb{P}\, 2(\hat{\mu}_{\cdot s0}(W) - \hat{\mu}_{\cdot 10}(W)) \times \Big( (\mu_{\cdot 0_{-s}0}(Z_s, W) - \hat{\mu}_{\cdot 0_{-s}0}(Z_s, W))\pi_{1s0}(W, Z_s)\frac{\mathbb{1}\{D=0\}}{p(D=0)}$$

$$+ (\hat{\mu}_{\cdot 0_{-s}0}(Z_s, W) - \hat{\mu}_{\cdot s0}(W))\frac{\mathbb{1}\{D=1\}}{p(D=1)}$$

$$- (\mu_{\cdot\cdot 0}(W, Z) - \hat{\mu}_{\cdot\cdot 0}(W, Z))\pi_{110}(W, Z)\frac{\mathbb{1}\{D=0\}}{p(D=0)}$$

$$- (\hat{\mu}_{\cdot\cdot 0}(W, Z) - \hat{\mu}_{\cdot 10}(W))\frac{\mathbb{1}\{D=1\}}{p(D=1)}\Big)$$

$$\tag{38}$$

$$+ \mathbb{P}((\hat{\mu}_{\cdot s0}(W) - \hat{\mu}_{\cdot 10}(W))^2 - (\mu_{\cdot s0}(W) - \mu_{\cdot 10}(W))^2)\frac{\mathbb{1}\{D=1\}}{p(D=1)} \tag{39}$$

$$+ o_p(n^{-1/2}), \tag{40}$$

Given the true density ratios, we can further simplify the expectations over $D = 0$ weighted by the density ratios in the expression above to expectations over $D = 1$. By definition of $\mu_{\cdot 0_{-s}0}(Z_s, W)$ in

(18) and $\mu_{\cdot s0}(W)$ in (20) and the definition of $\mu_{\cdot \cdot 0}(W, Z)$ and $\mu_{\cdot 10}(W)$ in Section F.1, (33) simplifies to

$$
\begin{aligned}
(33) = & \mathbb{P}_1 \, 2(\hat{\mu}_{\cdot s0}(W) - \hat{\mu}_{\cdot 10}(W))(\mu_{\cdot s0}(W) - \hat{\mu}_{\cdot s0}(W)) \\
& - \mathbb{P}_1 \, 2(\hat{\mu}_{\cdot s0}(W) - \hat{\mu}_{\cdot 10}(W))(\mu_{\cdot 10}(W) - \hat{\mu}_{\cdot 10}(W)) \\
& + \mathbb{P}_1(\hat{\mu}_{\cdot s0}(W) - \hat{\mu}_{\cdot 10}(W))^2 - (\mu_{\cdot s0}(W) - \mu_{\cdot 10}(W))^2) \\
& + o_p(n^{-1/2}),
\end{aligned}
$$

which is $o_p(n^{-1/2})$ as long as the convergence conditions in Condition F.2 hold.

As the denominator $v_{\mathsf{Z}}^{\mathrm{den}}$ is equal to the numerator $v_{\mathsf{Z}}^{\mathrm{num}}(\emptyset)$, it follows that the one-step estimator for the denominator $\hat{v}_{\mathsf{Z}}^{\mathrm{den}}$ is asymptotically linear with influence function $\psi_{\mathsf{Z},\emptyset}^{\mathrm{num}}$. $\qquad\square$

*Proof for Theorem F.3.* Combining Lemma G.2 and the Delta method [46, Theorem 3.1], the estimator $\hat{v}_{\mathsf{Z}}(s) = \hat{v}_{\mathsf{Z}}^{\mathrm{num}}(s)/\hat{v}_{\mathsf{Z}}^{\mathrm{den}}$ is asymptotically linear

$$
\frac{\hat{v}_{\mathsf{Z}}^{\mathrm{num}}(s)}{\hat{v}_{\mathsf{Z}}^{\mathrm{den}}} - \frac{v_{\mathsf{Z}}^{\mathrm{num}}(s)}{v_{\mathsf{Z}}^{\mathrm{den}}} = \mathbb{P}_n \psi_{\mathsf{Z},s}(D, W, Z, Y; \eta_{\mathsf{Z},s}^{\mathrm{num}}, \eta_{\mathsf{Z}}^{\mathrm{den}}) + o_p(n^{-1/2}),
$$

with influence function

$$
\psi_{\mathsf{Z},s}(D, W, Z, Y; \eta_{\mathsf{Z},s}^{\mathrm{num}}, \eta_{\mathsf{Z},s}^{\mathrm{den}}) = \frac{1}{v_{\mathsf{Z}}^{\mathrm{den}}} \psi_{\mathsf{Z},s}^{\mathrm{num}}(D, W, Z, Y; \eta_{\mathsf{Z},s}^{\mathrm{num}}) - \frac{v_{\mathsf{Z}}^{\mathrm{num}}(s)}{(v_{\mathsf{Z}}^{\mathrm{den}})^2} \psi_{\mathsf{Z}}^{\mathrm{den}}(D, W, Z, Y; \eta_{\mathsf{Z}}^{\mathrm{den}}),
$$

(41)

where $\psi_{\mathsf{Z},s}^{\mathrm{num}}(D, W, Z, Y; \eta_{\mathsf{Z},s}^{\mathrm{num}})$ is defined in (28) and $\psi_{\mathsf{Z}}^{\mathrm{den}}(D, W, Z, Y; \eta_{\mathsf{Z}}^{\mathrm{den}}) = \psi_{\mathsf{Z},\emptyset}^{\mathrm{num}}(D, W, Z, Y; \eta_{\mathsf{Z},\emptyset}^{\mathrm{num}})$.

Accordingly, the estimator asymptotically follows the normal distribution,

$$
\sqrt{n}\,(\hat{v}_{\mathsf{Z}}(s) - v_{\mathsf{Z}}(s)) \to_d N(0, \mathrm{var}(\psi_{\mathsf{Z},s}(D, W, Z, Y; \eta_{\mathsf{Z},s}^{\mathrm{num}}, \eta_{\mathsf{Z},s}^{\mathrm{den}}))) \tag{42}
$$

$\qquad\square$

### G.3 Value of $s$-partial conditional outcome shifts

Let the nuisance parameters in $v_{\mathsf{Y},\mathrm{bin}}^{\mathrm{num}}$ be denoted $\eta_{\mathsf{Y},s}^{\mathrm{num}} = (Q_{\mathrm{bin}}, \mu_{\cdot\cdot 1}, \mu_{\cdot\cdot s}, \pi)$ and its estimate as $\hat{\eta}_{\mathsf{Y},s}^{\mathrm{num}}$.

We represent the one-step corrected estimator for $v_{\mathsf{Y},\mathrm{bin}}^{\mathrm{num}}(s)$ as the V-statistic

$$
\begin{aligned}
\hat{v}_{\mathsf{Y},\mathrm{bin}}^{\mathrm{num}}(s) = & \mathbb{P}_{1,n} \tilde{\mathbb{P}}_{1,n} \left( \hat{\mu}_{\cdot\cdot 1}(W, Z) - \hat{\mu}_{\cdot\cdot s}(W, Z) \right)^2 & (43) \\
& + 2\mathbb{P}_{1,n} \tilde{\mathbb{P}}_{1,n} \left( \hat{\mu}_{\cdot\cdot 1}(W, Z) - \hat{\mu}_{\cdot\cdot s}(W, Z) \right) \left( \ell - \mu_{\cdot\cdot 1}(W, Z) \right) & (44) \\
& - 2\mathbb{P}_{1,n} \tilde{\mathbb{P}}_{1,n} \Big[ \left( \hat{\mu}_{\cdot\cdot 1}(W, Z_s, \tilde{Z}_{-s}) - \hat{\mu}_{\cdot\cdot s}(W, Z_s, \tilde{Z}_{-s}) \right) & \\
& \qquad\qquad (\ell(W, Z_s, \tilde{Z}_{-s}, Y) - \mu_{\cdot\cdot s}(W, Z_s, \tilde{Z}_{-s}))\pi(\tilde{Z}_{-s}, Z_s, W, q_{\mathrm{bin}}(W, Z)) \Big] & (45) \\
= & \mathbb{P}_{1,n} \tilde{\mathbb{P}}_{1,n} h\left( W, Z, Y, \tilde{W}, \tilde{Z}, \tilde{Y}; \hat{\eta}_{\mathsf{Y},s}^{\mathrm{num}} \right). & (46)
\end{aligned}
$$

In more detail, the conditions in Theorem 3.2 are as follows.

**Condition G.3.** *For variable subset $s$, suppose the following hold*

- $\pi(W, Z_s, Z_{-s}, Q_{bin})$ *is bounded*

- $\hat{\pi}$ *is consistent*

- $\mathbb{P}_1 \left( \hat{\mu}_{\cdot\cdot 0} - \mu_{\cdot\cdot 0} \right)^2 = o_p(n^{-1/2})$

- $\mathbb{P}_1 \left( \hat{\mu}_{\cdot\cdot 1} - \mu_{\cdot\cdot 1} \right)^2 = o_p(n^{-1/2})$

- $\mathbb{P}_1 \left( \hat{\mu}_{\cdot\cdot s} - \mu_{\cdot\cdot s} \right)^2 = o_p(n^{-1/2})$

- $\mathbb{P}_1 \left( \hat{q}_{bin} - q_{bin} \right)^2 = o_p(n^{-1})$

- $\mathbb{P}_1 \left( \hat{\mu}_{\cdot \cdot s} - \mu_{\cdot \cdot s} \right) \left( \hat{\pi} - \pi \right) = o_p(n^{-1/2})$.

**Lemma G.4.** *Assuming Condition G.3 holds, $\hat{v}^{num}_{Y,bin}$ is an asymptotically linear estimator for $v^{num}_{Y,bin}$, i.e.*

$$\hat{v}^{num}_{Y,bin}(s) - v^{num}_{Y,bin}(s) = \mathbb{P}_{1,n} \psi^{num}_{Y,s}(D, W, Z, Y; \eta^{num}_{Y,s}) + o_p(n^{-1/2}), \tag{47}$$

*with influence function*

$$
\begin{aligned}
\psi^{num}_{Y,s} \left( d, w, z, y; \eta^{num}_{Y,s} \right) =& (\mu_{\cdot \cdot 1}(w,z) - \mu_{\cdot \cdot s}(w,z))^2 \\
&+ 2(\mu_{\cdot \cdot 1}(w,z) - \mu_{\cdot \cdot s}(w,z)) \left[ \ell(w,z,y) - \mu_{\cdot \cdot 1}(w,z) \right] \\
&- 2\mathbb{P}_1 \Big[ \left( \mu_{\cdot \cdot 1}(w, z_s, Z_{-s}) - \mu_{\cdot \cdot s}(w, z_s, Z_{-s}) \right) \\
&\qquad\quad \left[ \ell \left( w, z_s, Z_{-s}, y \right) - \mu_{\cdot \cdot s}(w, z_s, Z_{-s}) \right] \pi \left( Z_{-s}, z_s, w, q_{bin}(w,z) \right) \Big] \\
&- v^{num}_{Y,bin}(s).
\end{aligned}
\tag{48}
$$

*Proof.* Defining the symmetrized version of $h$ in (46) as $h_{sym}(W, Z, Y, \tilde{W}, \tilde{Z}, \tilde{Y}) = \frac{h(W,Z,Y,\tilde{W},\tilde{Z},\tilde{Y}) + h(\tilde{W},\tilde{Z},\tilde{Y},W,Z,Y)}{2}$, we rewrite the estimator as

$$\hat{v}^{num}_{Y,bin}(s) = \mathbb{P}_{1,n} \tilde{\mathbb{P}}_{1,n} h_{sym} \left( W, Z, Y, \tilde{W}, \tilde{Z}, \tilde{Y}; \hat{\eta}^{num}_{Y,s} \right).$$

Per Theorem 12.3 in [46], the Hájek projection of $\hat{v}^{num}_{Y,bin}(s)$ is

$$
\begin{aligned}
\hat{u}^{num}_{Y,bin}(s) &= \sum_{i=1}^{n} \mathbb{P}_1 \left[ \mathbb{P}_{1,n} \tilde{\mathbb{P}}_{1,n} h_{sym} \left( W, Z, Y, \tilde{W}, \tilde{Z}, \tilde{Y}; \hat{\eta}^{num}_{Y,s} \right) - \bar{v}^{num}_Y(s) \mid X_i, Y_i \right] \\
&= \sum_{i=1}^{n} \mathbb{P}_1 \left[ h_{sym} \left( X_i, Y_i, X^{(2)}, Y^{(2)}; \hat{\eta}^{num}_{Y,s} \right) - \bar{v}^{num}_Y(s) \mid X_i, Y_i \right] \\
&= \sum_{i=1}^{n} h_{sym,1} \left( X_i, Y_i; \hat{\eta}^{num}_{Y,s} \right)
\end{aligned}
$$

where $\bar{\bar{v}}^{num}_Y(s) = \mathbb{P}_1 \tilde{\mathbb{P}}_1 h_{sym} \left( W, Z, Y, \tilde{W}, \tilde{Z}, \tilde{Y}; \hat{\eta}^{num}_{Y,s} \right)$.

Consider the decomposition

$$
\begin{aligned}
\hat{v}^{num}_{Y,bin}(s) - v^{num}_{Y,bin}(s) =& \mathbb{P}_{1,n} \tilde{\mathbb{P}}_{1,n} h_{sym} \left( W, Z, Y, \tilde{W}, \tilde{Z}, \tilde{Y}; \hat{\eta}^{num}_{Y,s} \right) - \mathbb{P}_1 \tilde{\mathbb{P}}_1 h_{sym} \left( W, Z, Y, \tilde{W}, \tilde{Z}, \tilde{Y}; \eta^{num}_{Y,s} \right) \\
=& \mathbb{P}_{1,n} \tilde{\mathbb{P}}_{1,n} h_{sym} \left( W, Z, Y, \tilde{W}, \tilde{Z}, \tilde{Y}; \hat{\eta}^{num}_{Y,s} \right) - \mathbb{P}_{1,n} \left[ h_{sym,1} \left( X, Y; \hat{\eta}^{num}_{Y,s} \right) + \bar{\bar{v}}^{num}_Y(s) \right]
\end{aligned}
\tag{49}
$$

$$+ (\mathbb{P}_{1,n} - \mathbb{P}_1) \left( h_{sym,1} \left( X, Y; \hat{\eta}^{num}_{Y,s} \right) + \bar{\bar{v}}^{num}_Y(s) - h_{sym,1} \left( X, Y \right) - v^{num}_Y(s) \right) \tag{50}$$

$$+ (\mathbb{P}_{1,n} - \mathbb{P}_1) \left( h_{sym,1} \left( X, Y \right) + v^{num}_Y(s) \right) \tag{51}$$

$$+ \mathbb{P}_1 \left( h_{sym,1} \left( X, Y; \hat{\eta}^{num}_{Y,s} \right) + \bar{\bar{v}}^{num}_Y(s) - h_{sym,1} \left( X, Y; \eta^{num}_{Y,s} \right) - v^{num}_Y(s) \right). \tag{52}$$

We analyze each term in turn.

Term (49): Suppose $\mathbb{P}_1 h^2_{sym}(W, Z, Y, \tilde{W}, \tilde{Z}, \tilde{Y}; \hat{\eta}^{num}_{Y,s}) < \infty$. Via a straightforward extension of the proof in Theorem 12.3 in van der Vaart [46], one can show that

$$\frac{var \left( \mathbb{P}_{1,n} \tilde{\mathbb{P}}_{1,n} h_{sym} \left( W, Z, Y, \tilde{W}, \tilde{Z}, \tilde{Y}; \hat{\eta}^{num}_{Y,s} \right) \right)}{var \left( \mathbb{P}_{1,n} h_{sym,1} \left( W, Z, Y; \hat{\eta}^{num}_{Y,s} \right) \right)} \to_p 1.$$

Then by Theorem 11.2 in van der Vaart [46] and Slutsky's lemma, we have

$$\mathbb{P}_{1,n}\tilde{\mathbb{P}}_{1,n}h_{sym}\left(W,Z,Y,\tilde{W},\tilde{Z},\tilde{Y};\hat{\eta}^{\texttt{num}}_{\texttt{Y},s}\right)-\mathbb{P}_{1,n}\left[h_{sym,1}\left(W,Z,Y;\hat{\eta}^{\texttt{num}}_{\texttt{Y},s}\right)+\bar{\hat{v}}^{\texttt{num}}_{\texttt{Y}}(s)\right]=o_p\left(n^{-1/2}\right).$$

Term (50): We perform sample splitting to estimate the nuisance parameters and calculate the estimator for $\hat{v}^{\texttt{num}}_{\texttt{Y}}(s)$. Then by Lemma 1 in Kennedy [25], we have that

$$(\mathbb{P}_{1,n}-\mathbb{P}_1)\left(h_{sym,1}\left(W,Z,Y;\hat{\eta}^{\texttt{num}}_{\texttt{Y},s}\right)+\bar{\hat{v}}^{\texttt{num}}_{\texttt{Y}}(s)-h_{sym,1}\left(W,Z,Y;\eta^{\texttt{num}}_{\texttt{Y},s}\right)-v^{\texttt{num}}_{\texttt{Y}}(s)\right)=o_p(n^{-1/2})$$

as long as the estimators for the nuisance parameters are consistent.

Term (51): This term $(\mathbb{P}_{1,n}-\mathbb{P}_1)\left(h_{sym,1}\left(W,Z,Y;\eta^{\texttt{num}}_{\texttt{Y},s}\right)+v^{\texttt{num}}_{\texttt{Y}}(s)\right)=$ $(\mathbb{P}_{1,n}-\mathbb{P}_1)h_{sym,1}\left(W,Z,Y;\eta^{\texttt{num}}_{\texttt{Y},s}\right)$ follows an asymptotic normal distribution per CLT.

Term (52): We will show that this bias term is asymptotically negligible. For notational simplicity, let $\hat{\xi}(W,Z_s,Z_{-s})=\hat{\mu}_{\cdot\cdot 1}(W,Z)-\hat{\mu}_{\cdot\cdot s}(W,Z)$.

$$\mathbb{P}_1\tilde{\mathbb{P}}_1\left(h_{sym}\left(W,Z,Y,\tilde{W},\tilde{Z},\tilde{Y};\hat{\eta}^{\texttt{num}}_{\texttt{Y},s}\right)+\bar{\hat{v}}^{\texttt{num}}_{\texttt{Y}}(s)-h_{sym}\left(W,Z,Y,\tilde{W},\tilde{Z},\tilde{Y};\eta^{\texttt{num}}_{\texttt{Y},s}\right)-v^{\texttt{num}}_{\texttt{Y}}(s)\right)$$

$$=\mathbb{P}_1\tilde{\mathbb{P}}_1\left(h\left(W,Z,Y,\tilde{W},\tilde{Z},\tilde{Y};\hat{\eta}^{\texttt{num}}_{\texttt{Y},s}\right)-h\left(W,Z,Y,\tilde{W},\tilde{Z},\tilde{Y};\eta^{\texttt{num}}_{\texttt{Y},s}\right)\right)$$

$$=\mathbb{P}_1\left(\hat{\mu}_{\cdot\cdot 1}(W,Z)-\hat{\mu}_{\cdot\cdot s}(W,Z)\right)^2-\mathbb{P}_1\left(\mu_{\cdot\cdot 1}(W,Z)-\mu_{\cdot\cdot s}(W,Z)\right)^2$$
$$+2\mathbb{P}_1\left(\hat{\mu}_{\cdot\cdot 1}(W,Z)-\hat{\mu}_{\cdot\cdot s}(W,Z)\right)\left[\mu_{\cdot\cdot 1}(W,Z)-\hat{\mu}_{\cdot\cdot 1}(W,Z)\right]$$
$$-2\mathbb{P}_1\tilde{\mathbb{P}}_1\Big\{\left(\hat{\mu}_{\cdot\cdot 1}(W,Z_s,\tilde{Z}_{-s})-\hat{\mu}_{\cdot\cdot s}(W,Z_s,\tilde{Z}_{-s})\right)\left[\ell(W,Z_s,\tilde{Z}_{-s},Y)-\hat{\mu}_{\cdot\cdot s}(W,Z_s,\tilde{Z}_{-s})\right]$$
$$\hat{\pi}\left(\tilde{Z}_{-s},Z_s,W,\hat{q}_{\texttt{bin}}(W,Z)\right)\Big\}$$

$$=\mathbb{P}_1\left(\hat{\mu}_{\cdot\cdot 1}(W,Z)-\hat{\mu}_{\cdot\cdot s}(W,Z)\right)^2-\mathbb{P}_1\left(\mu_{\cdot\cdot 1}(W,Z)-\mu_{\cdot\cdot s}(W,Z)\right)^2$$
$$+2\mathbb{P}_1\left(\hat{\mu}_{\cdot\cdot 1}(W,Z)-\hat{\mu}_{\cdot\cdot s}(W,Z)\right)\left[\mu_{\cdot\cdot 1}(W,Z)-\hat{\mu}_{\cdot\cdot 1}(W,Z)\right]$$
$$-2\mathbb{P}_1\tilde{\mathbb{P}}_1\hat{\xi}(W,Z_s,\tilde{Z}_{-s})\left[\mu_{\cdot\cdot s}(W,Z_s,\tilde{Z}_{-s})-\hat{\mu}_{\cdot\cdot s}(W,Z_s,\tilde{Z}_{-s})\right]\hat{\pi}\left(\tilde{Z}_{-s},Z_s,W,q_{\texttt{bin}}(W,Z)\right)$$
$$-2\mathbb{P}_1\tilde{\mathbb{P}}_1\Big\{\hat{\xi}(W,Z_s,\tilde{Z}_{-s})\left[\ell(W,Z_s,\tilde{Z}_{-s},Y)-\hat{\mu}_{\cdot\cdot s}(W,Z_s,\tilde{Z}_{-s})\right]$$
$$\left[\hat{\pi}\left(\tilde{Z}_{-s},Z_s,W,\hat{q}_{\texttt{bin}}(W,Z)\right)-\hat{\pi}\left(\tilde{Z}_{-s},Z_s,W,q_{\texttt{bin}}(W,Z)\right)\right]\Big\}$$

$$=\mathbb{P}_1\left(\mu_{\cdot\cdot s}(W,Z)-\hat{\mu}_{\cdot\cdot s}(W,Z)\right)\left(\hat{\mu}_{\cdot\cdot 1}(W,Z)-\hat{\mu}_{\cdot\cdot s}(W,Z)+\mu_{\cdot\cdot 1}(W,Z)-\mu_{\cdot\cdot s}(W,Z)\right) \quad (53)$$
$$+\mathbb{P}_1\left(\hat{\mu}_{\cdot\cdot 1}(W,Z)-\mu_{\cdot\cdot 1}(W,Z)\right)\left(\mu_{\cdot\cdot 1}(W,Z)-\mu_{\cdot\cdot s}(W,Z)-\hat{\mu}_{\cdot\cdot 1}(W,Z)+\hat{\mu}_{\cdot\cdot s}(W,Z)\right) \quad (54)$$
$$-2\mathbb{P}_1\tilde{\mathbb{P}}_1\hat{\xi}(W,Z_s,\tilde{Z}_{-s})\left[\mu_{\cdot\cdot s}(W,Z_s,\tilde{Z}_{-s})-\hat{\mu}_{\cdot\cdot s}(W,Z_s,\tilde{Z}_{-s})\right]\hat{\pi}\left(\tilde{Z}_{-s},Z_s,W,q_{\texttt{bin}}(W,Z)\right)$$
$$\quad (55)$$
$$-2\mathbb{P}_1\tilde{\mathbb{P}}_1\Big\{\hat{\xi}(W,Z_s,\tilde{Z}_{-s})\left[\ell(W,Z_s,\tilde{Z}_{-s},Y)-\hat{\mu}_{\cdot\cdot s}(W,Z_s,\tilde{Z}_{-s})\right]$$
$$\left[\hat{\pi}\left(\tilde{Z}_{-s},Z_s,W,\hat{q}_{\texttt{bin}}(W,Z)\right)-\hat{\pi}\left(\tilde{Z}_{-s},Z_s,W,q_{\texttt{bin}}(W,Z)\right)\right]\Big\}. \quad (56)$$

Note that (56) is $o_p(n^{-1/2})$ under the assumed convergence rates for $\hat{q}_{\texttt{bin}}$. In addition, (54) is $o_p(n^{-1/2})$, under the assumed convergence rates for $\hat{\mu}_{\cdot\cdot 1}$ and $\hat{\mu}_{\cdot\cdot s}$.

Analyzing the remaining summands (53) + (55), we note that it simplifies as follows:

$$\mathbb{P}_1\left(\mu_{..s}(X) - \hat{\mu}_{..s}(X)\right)\left(\hat{\mu}_{..1}(X) - \hat{\mu}_{..s}(X) + \mu_{..1}(X) - \mu_{..s}(X)\right)$$
$$- 2\mathbb{P}_1\tilde{\mathbb{P}}_1\Big\{\hat{\xi}(W, Z_s, \tilde{Z}_{-s})\left[\mu_{..s}(W, Z_s, \tilde{Z}_{-s}) - \hat{\mu}_{..s}(W, Z_s, \tilde{Z}_{-s})\right]$$
$$\left(\hat{\pi}\left(\tilde{Z}_{-s}, Z_s, W, q_{\mathtt{bin}}(W, Z)\right)\right) - \pi\left(\tilde{Z}_{-s}, Z_s, W, q_{\mathtt{bin}}(W, Z)\right)\Big)\Big\}$$
$$- 2\mathbb{P}_1\tilde{\mathbb{P}}_1\hat{\xi}(W, Z_s, \tilde{Z}_{-s})\left[\mu_{..s}(W, Z_s, \tilde{Z}_{-s}) - \hat{\mu}_{..s}(W, Z_s, \tilde{Z}_{-s})\right]\pi\left(\tilde{Z}_{-s}, Z_s, W, q_{\mathtt{bin}}(W, Z)\right)\Big)$$
$$=\mathbb{P}_1\left(\mu_{..s}(X) - \hat{\mu}_{..s}(X)\right)\left(\mu_{..1}(X) - \hat{\mu}_{..1}(X) - \mu_{..s}(X) + \hat{\mu}_{..s}(X)\right)$$
$$- 2\mathbb{P}_1\tilde{\mathbb{P}}_1\Big\{\hat{\xi}(W, Z_s, \tilde{Z}_{-s})\left[\mu_{..s}(W, Z_s, \tilde{Z}_{-s}) - \hat{\mu}_{..s}(W, Z_s, \tilde{Z}_{-s})\right]$$
$$\left[\hat{\pi}\left(\tilde{Z}_{-s}, Z_s, W, q_{\mathtt{bin}}(W, Z)\right)\right) - \pi\left(\tilde{Z}_{-s}, Z_s, W, q_{\mathtt{bin}}(W, Z)\right)\Big]\Big\},$$

which is $o_p(n^{-1/2})$, under the assumed convergence rates for $\hat{\mu}_{..s}$, $\hat{\mu}_{..1}$, and $\hat{\pi}$. $\qquad\square$

**Condition G.5** (Convergence conditions for $\hat{v}_{\mathtt{Y}}^{\mathtt{den}}$). *Suppose the following holds*

- $\mathbb{P}_1(\mu_{..1} - \mu_{..0} - (\hat{\mu}_{..1} - \hat{\mu}_{..0}))^2 = o_p(n^{-1/2})$

- $\mathbb{P}_0(\mu_{..0} - \hat{\mu}_{..0})(\pi_{110} - \hat{\pi}_{110}) = o_p(n^{-1/2})$

- $\mathbb{P}_0(\mu_{..1} - \mu_{..0})^2$ *is bounded*

- *(Positivity)* $p(w, z|d = 0) > 0$ *almost everywhere, such that the density ratios* $\pi_{110}(w, z)$ *are well-defined and bounded.*

Let the nuisance parameters in the one-step estimator $v_{\mathtt{Y}}^{\mathtt{den}}$ be denoted by $\eta_{\mathtt{Y}}^{\mathtt{den}} = (\mu_{..0}, \mu_{..1}, \pi_{110})$ and the set of estimated nuisances by $\hat{\eta}_{\mathtt{Y}}^{\mathtt{den}}$.

**Lemma G.6.** *Assuming Condition G.5 holds, then $\hat{v}_{\mathtt{Y}}^{den}$ is an asymptotically linear estimator for $v_{\mathtt{Y}}^{den}$, i.e.*

$$\hat{v}_{\mathtt{Y}}^{den} - v_{\mathtt{Y}}^{den} = \mathbb{P}_n\psi_{\mathtt{Y}}^{den}(D, W, Z, Y; \eta_{\mathtt{Y}}^{den}) + o_p(n^{-1/2})$$

*with influence function*

$$\psi_{\mathtt{Y}}^{den}(D, W, Z, Y; \eta_{\mathtt{Y}}^{den}) = (\mu_{..1}(W, Z) - \mu_{..0}(W, Z))^2 \frac{\mathbb{1}\{D = 1\}}{p(D = 1)} \tag{57}$$

$$+ 2(\mu_{..1}(W, Z) - \mu_{..0}(W, Z))(\ell - \mu_{..1}(W, Z))\frac{\mathbb{1}\{D = 1\}}{p(D = 1)} \tag{58}$$

$$- 2(\mu_{..1}(W, Z) - \mu_{..0}(W, Z))(\ell - \mu_{..0}(W, Z))\pi_{110}(W, Z)\frac{\mathbb{1}\{D = 0\}}{p(D = 0)} \tag{59}$$

$$- v_{\mathtt{Y}}^{den}. \tag{60}$$

*Proof.* Consider the following decomposition of bias in the one-step corrected estimate

$$\hat{v}_{\mathtt{Y}}^{\mathtt{den}} - v_{\mathtt{Y}}^{\mathtt{den}}$$
$$=(\mathbb{P}_n - \mathbb{P})\psi_{\mathtt{Y}}^{\mathtt{den}}(D, W, Z, Y; \eta_{\mathtt{Y}}^{\mathtt{den}}) \tag{61}$$
$$+ (\mathbb{P}_n - \mathbb{P})(\psi_{\mathtt{Y}}^{\mathtt{den}}(D, W, Z, Y; \hat{\eta}_{\mathtt{Y}}^{\mathtt{den}}) - \psi_{\mathtt{Y}}^{\mathtt{den}}(D, W, Z, Y; \eta_{\mathtt{Y}}^{\mathtt{den}})) \tag{62}$$
$$+ \mathbb{P}(\psi_{\mathtt{Y}}^{\mathtt{den}}(D, W, Z, Y; \hat{\eta}_{\mathtt{Y}}^{\mathtt{den}}) - \psi_{\mathtt{Y}}^{\mathtt{den}}(D, W, Z, Y; \eta_{\mathtt{Y}}^{\mathtt{den}})) \tag{63}$$

We observe that (61) converges to a normal distribution per CLT assuming that the variance of $\psi_{\mathtt{Y}}^{\mathtt{den}}$ is finite. The empirical process term (62) is asymptotically negligible since the nuisance parameters $\eta_{\mathtt{Y}}^{\mathtt{den}}$ are evaluated on an separate evaluation data split from the training data used for estimation.

In addition assuming that the estimators for the nuisance parameters are consistent, Kennedy [25, Lemma 1] states that

$$(\mathbb{P}_n - \mathbb{P})(\psi_Y^{\text{den}}(D, W, Z, Y; \hat{\eta}_Y^{\text{den}}) - \psi_Y^{\text{den}}(D, W, Z, Y; \eta_Y^{\text{den}})) = o_p(n^{-1/2}).$$

We now show that the remainder term (63) is also asymptotically negligible. Substituting the influence function and integrating with respect to $Y$, (63) becomes

$$(63) = \mathbb{P} \left( \hat{\mu}_{..1}(W, Z) - \hat{\mu}_{..0}(W, Z) - (\mu_{..1}(W, Z) - \mu_{..0}(W, Z)) \right)^2 \frac{\mathbb{1}(D = 1)}{p(D = 1)} \tag{64}$$

$$+ 2\mathbb{P} \left( \hat{\mu}_{..1}(W, Z) - \hat{\mu}_{..0}(W, Z) - (\mu_{..1}(W, Z) - \mu_{..0}(W, Z)) \right) (\mu_{..1}(W, Z) - \mu_{..0}(W, Z)) \frac{\mathbb{1}(D = 1)}{p(D = 1)} \tag{65}$$

$$+ 2\mathbb{P} \left( \hat{\mu}_{..1}(W, Z) - \hat{\mu}_{..0}(W, Z) \right) (\mu_{..1}(W, Z) - \hat{\mu}_{..1}(W, Z)) \frac{\mathbb{1}\{D = 1\}}{p(D = 1)} \tag{66}$$

$$- 2\mathbb{P} \left( \hat{\mu}_{..1}(W, Z) - \hat{\mu}_{..0}(W, Z) \right) (\mu_{..0}(W, Z) - \hat{\mu}_{..0}(W, Z)) \hat{\pi}_{110}(W, Z) \frac{\mathbb{1}\{D = 0\}}{p(D = 0)} \tag{67}$$

$$= \mathbb{P} \left( \hat{\mu}_{..1}(W, Z) - \hat{\mu}_{..0}(W, Z) - (\mu_{..1}(W, Z) - \mu_{..0}(W, Z)) \right)^2 \frac{\mathbb{1}(D = 1)}{p(D = 1)} \tag{68}$$

$$+ 2\mathbb{P}(\mu_{..0}(W, Z) - \hat{\mu}_{..0}(W, Z))(\mu_{..1}(W, Z) - \mu_{..0}(W, Z)) \frac{\mathbb{1}(D = 1)}{p(D = 1)} \tag{69}$$

$$- 2\mathbb{P} \left( \hat{\mu}_{..1}(W, Z) - \hat{\mu}_{..0}(W, Z) \right) (\mu_{..0}(W, Z) - \hat{\mu}_{..0}(W, Z))(\hat{\pi}_{110}(W, Z) - \pi_{110}(W, Z)) \frac{\mathbb{1}\{D = 0\}}{p(D = 1)} \tag{70}$$

$$- 2\mathbb{P} \left( \hat{\mu}_{..1}(W, Z) - \hat{\mu}_{..0}(W, Z) \right) (\mu_{..0}(W, Z) - \hat{\mu}_{..0}(W, Z)) \pi_{110}(W, Z) \frac{\mathbb{1}\{D = 0\}}{p(D = 0)} \tag{71}$$

$$= \mathbb{P} \left( \hat{\mu}_{..1}(W, Z) - \hat{\mu}_{..0}(W, Z) - (\mu_{..1}(W, Z) - \mu_{..0}(W, Z)) \right)^2 \frac{\mathbb{1}(D = 1)}{p(D = 1)} \tag{72}$$

$$- 2\mathbb{P} \left( \hat{\mu}_{..1}(W, Z) - \hat{\mu}_{..0}(W, Z) \right) (\mu_{..0}(W, Z) - \hat{\mu}_{..0}(W, Z))(\hat{\pi}_{110}(W, Z) - \pi_{110}(W, Z)) \frac{\mathbb{1}\{D = 0\}}{p(D = 0)} \tag{73}$$

Thus the remainder term is $o_p(n^{-1/2})$ if Condition G.5 holds. $\qquad \square$

*Proof for Theorem 3.2.* Combining Lemmas G.4, G.6, and the Delta method [46, Theorem 3.1], the estimator $\hat{v}_{Y,\text{bin}}(s) = \hat{v}_{Y,\text{bin}}^{\text{num}}(s)/\hat{v}_Y^{\text{den}}$ is asymptotically linear

$$\frac{\hat{v}_{Y,\text{bin}}^{\text{num}}(s)}{\hat{v}_Y^{\text{den}}} - \frac{v_{Y,\text{bin}}^{\text{num}}(s)}{v_Y^{\text{den}}} = \mathbb{P}_n \psi_{Y,\text{bin},s}(D, W, Z, Y; \eta_{Y,s}^{\text{num}}, \eta_Y^{\text{den}}) + o_p(n^{-1/2}),$$

with influence function

$$\psi_{Y,\text{bin},s}(D, W, Z, Y; \eta_{Y,s}^{\text{num}}, \eta_Y^{\text{den}}) = \frac{1}{v_Y^{\text{den}}} \psi_{Y,s}^{\text{num}}(D, W, Z, Y; \eta_{Y,s}^{\text{num}}) - \frac{v_{Y,\text{bin}}^{\text{num}}(s)}{(v_Y^{\text{den}})^2} \psi_Y^{\text{den}}(D, W, Z, Y; \eta_Y^{\text{den}}), \tag{74}$$

where $\psi_{Y,s}^{\text{num}}$ and $\psi_Y^{\text{den}}$ are defined in (48) and (60).

Accordingly, the estimator follows a normal distribution asymptotically,

$$\sqrt{n} \left( \hat{v}_Y(s) - v_Y(s) \right) \to_d N(0, \text{var}(\psi_{Y,\text{bin},s}(D, W, Z, Y; \eta_Y^{\text{num}}, \eta_Y^{\text{den}}))) \tag{75}$$

$\qquad \square$

*Remark on super-fast convergence of $\hat{q}_{\text{bin}}$.* One of the conditions in Condition G.3 states that the $\hat{q}_{\text{bin}}$ converges at $o_p(n^{-1})$ rate. While this seems restrictive, binned risk converges exponentially under suitable margin conditions presented in Audibert and Tsybakov [2]. In particular, consider the following margin condition which is a relaxation of Condition 3.1.

**Condition G.7.** *For all bin edges $b$ except $b \in \{0, 1\}$, the set $\Xi_\epsilon = \{x : |q(x) - b| \leq \epsilon\}$ is measure zero for some $\epsilon > 0$.*

Suppose the function $q$ belongs to the Hölder class $\Sigma(\beta, L, \mathbb{R}^d)$ for positive integer $\beta$ and $L > 0$ (see definition in Audibert and Tsybakov [2]), the marginal law of $X$ satisfies the strong density assumption, and the margin condition G.7 is satisfied. Then the rate of convergence of $\hat{q}_{\texttt{bin}}$ is exponential, i.e.

$$\mathbb{E}\left(\hat{q}_{\texttt{bin}}(X) - q_{\texttt{bin}}(X)\right)^2 \leq C_1 \exp(-C_2 n) \tag{76}$$

for constants $C_1, C_2 > 0$ that do not depend on $n$.

*Remark on value of $s$-partial conditional outcome shift.* Unlike the case of $s$-partial conditional covariate shift, note that $v_{\mathtt{Y}}(\emptyset)$ may be non-zero when the risk in the target domain is a recalibration (i.e. temperature-scaling) of the risk in the source domain [34, 19]. For instance, there may be general environmental factors such that readmission risks in the target domain are uniformly lower.

## H    Implementation details

Here we describe how the nuisance parameters can be estimated in each of the decompositions. In general, density ratio models can be estimated via a standard reduction to a classification problem where a probabilistic classifier is trained to discriminate between source and target domains [43].

**Note on computation time**. Shapley value computation can be parallelized over the subsets. For high-dimensional tabular data, grouping together variables can further reduce computation time (and increase interpretability).

### H.1    Aggregate decompositions

**Density ratio models.** Using direct importance estimation [43], density ratio models $\pi_{100}(W)$ and $\pi_{110}(W, Z)$ can be estimated by fitting classifiers on the combined source and target data to predict $D = 0$ or $1$ from features $W$ and $(W, Z)$, respectively.

**Outcome models.** The outcome models $\mu_{.00}(W)$ and $\mu_{..0}(W, Z)$ can be fit in a number of ways. One option is to estimate the conditional distribution of the outcome (i.e. $p_0(Y|W)$ or $p_0(Y|W, Z)$) using binary classifiers, from which one can obtain an estimate of the conditional expectation of the loss. Alternatively, one can estimate the conditional expectations of the loss directly by fitting regression models.

### H.2    Detailed decomposition for $s$-partial outcome shift

**Density ratio models.** The density ratio $\pi(W, Z_s, Z_{-s}, Q) = p_1(Z_{-s}|W, Z_s, q(W, Z) = Q)/p_1(Z_{-s})$ in (6) can be estimated as follows. Create a second ("phantom") dataset of the target domain in which $Z_{-s}$ is independent of $Z_s$ by permuting the original $Z_{-s}$ in the target domain. Compute $q_{\texttt{bin}}$ for all observations in the original dataset and the permuted dataset. Concatenate the original dataset from the target domain with the permuted dataset. Train a classifier to predict if an observation is from the original versus the permuted dataset.

**Outcome models.** The outcome models $\mu_{..1}$ and $\mu_{..s}(W, Z)$ can be similarly fit by estimating the conditional distributions $p_0(Y|W, Z)$ and $p_s(Y|W, Z, q_{\texttt{bin}}(W, Z))$ on the target domain, and then taking expectation of the loss.

**Computing U-statistics.** Calculating the double average $\mathbb{P}_{1,n}\tilde{\mathbb{P}}_{1,n}$ in the estimator requires evaluating all $n^2$ pairs of data points in target domain. This can be computationally expensive, so a good approximation is to subsample the inner average. We take 2000 subsamples. We did not see large changes in the bias of the estimates compared to calculating the exact U-statistics.

### H.3    Detailed decomposition for $s$-partial conditional outcome shift

**Density ratio models.** The ratio $\pi_{1s0}(W, Z_s) = p_1(W, Z_s)/p_0(W, Z_s)$ can be similarly fit using direct importance estimation.

**Outcome models.** We require the following models.

- $\mu_{\cdot 0_{-s}0}(z_s, w) = \mathbb{E}_{\cdot 00}[\ell | z_s, w]$, defined in (18), can be estimated by regressing loss against $w, z_s$ on the source domain.

- $\mu_{\cdot 10}(w) = \mathbb{E}_{\cdot 10}[\ell | w]$, defined in (19), can be estimated by regressing $\mu_{\cdot \cdot 0}(w, z)$ against $w$ in the target domain.

- $\mu_{\cdot s0}(w) = \mathbb{E}_{\cdot s0}[\ell | z_s, w]$, defined in (20), can be estimated by regressing $\mu_{\cdot 0_{-s}0}(z_s, w)$ against $w$ in the target domain.

For all models, we use cross-validation to select among model types and hyperparameters. Model selection is important so that the convergence rate conditions for the asymptotic normality results are met.

## I   Simulation details

**Data generation**: We generate synthetic data under two settings. For the coverage checks in Section 4.1, all features are sampled independently from a multivariate normal distribution. The mean of the $(W, Z)$ in the source and target domains are $(0, 2, 0.7, 3)$ and $(0, 0, 0, 0)$, respectively. The outcome in the source and target domains are simulated from a logistic regression model with coefficients $(0.3, 1, 0.5, 1)$ and $(0.3, 0.1, 0.5, 1.4)$.

In the second setting for baseline comparisons in Figure 2b(ii), each feature in $W \in \mathbb{R}$ and $Z \in \mathbb{R}^5$ is sampled independently from the rest from a uniform distribution over $[-1, 1)$. The binary outcome $Y$ is sampled from a logistic regression model with coefficients $(0.2, 0.4, 2, 0.25, 0.1, 0.1)$ in source and $(0.2, -0.4, 0.8, 0.1, 0.1, 0.1)$ in target.

In both the settings, we analyze performance gap of logistic regression models fit on a held-out source dataset.

**Sample-splitting**: We fit all models on 80% of the data points from both source and target datasets which is the `Tr` partition, and keep the remaining 20% for computing the estimators which is the `Ev` partition.

**Model types**: We use `scikit-learn` implementations for all models [33]. We use 3-fold cross validation to select models. For density models, we fit random forest classifiers and logistic regression models with polynomial features of degree 3. We clip the predicted probabilities from the density model for $\pi$ at $10^{-6}$ to avoid very large density weights. Depending on whether the target outcome in outcome models is binary or real-valued, we fit random forest classifiers or regressors, and logistic regression or linear regression models with ridge penalty. Specific hyperparameter ranges for the grid search are provided in the code.

**Computing time and resources**: Computation for the VI estimates can be quite fast, as Shapley value computation can be parallelized over the subsets and the number of unique variable subsets sampled in the Shapley value approximation is often quite small. For instance, for the ACS Public Coverage case study with 34 features, the unique subsets is 131 even when the number of sampled subsets is 3000, and it takes around 160 seconds to estimate the value of a single variable subset. All experiments are run on a 2.60 GHz processor with 8 CPU cores.

## J   Data analysis details

**Synthetic.** We describe accuracy of the ML algorithm after it is retrained with the top $k$ features and predictions from the original model (Table 3). Proposed method results in the revised model with highest gain in accuracy and AUC. Figure 5 shows the aggregate and detailed decompositions for the simulation setup in Section 4.1. Figure 6 shows the coverage rates of 90% CIs for the aggregate decompositions.

**Hospital readmission.** Using data from the electronic health records of a large safety-net hospital in the US, we analyzed the transferability of performance measures of a Gradient Boosted Tree (GBT) trained to predict 30-day readmission risk for the general patient population (source) but applied to patients diagnosed with heart failure (target). Each of the source and target datasets have 3750 observations for analyzing the performance gap. The GBT is trained on a held-out sample of 18,873 points from the general population. Features include 4 demographic variables $(W)$ and 16 diagnosis

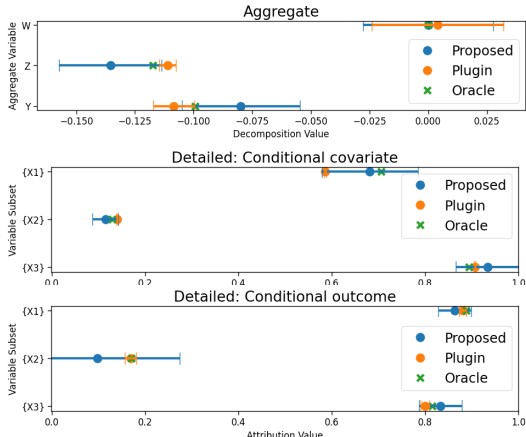

Figure 5: Sample estimates and CIs for simulation from Section 4.1. `Proposed` is debiased ML estimator for HDPD.

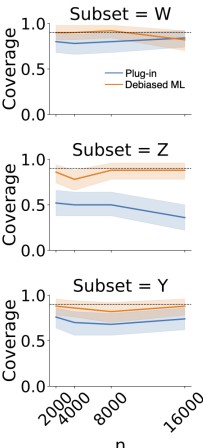

Figure 6: Coverage rates of 90% CIs for the aggregate decomposition terms for the simulation setup in Section 4.1.

Table 3: Accuracy and AUC for the revised model with respect to the top $k = \{1, 2, 3\}$ variables identified by different methods. Proposed method leads to a model with highest improvement in performance, thus, reducing the performance gap. We compare against two additional baselines that are outperformed by the proposed method: retraining a model on the target data with all features (AUC 0.89, Acc 0.84) and retraining on a weighted source-target data where loss for each point in source and target is weighted by a tuned parameter $\alpha$ and $1 - \alpha$ respectively (AUC 0.89, Acc 0.85).

| Method | AUC-1 | Acc-1 | AUC-2 | Acc-2 | AUC-3 | Acc-3 |
|---|---|---|---|---|---|---|
| ParametricChange | 0.87 | 0.82 | 0.87 | 0.82 | **0.91** | **0.86** |
| ParametricAcc | 0.87 | 0.82 | 0.87 | 0.82 | **0.91** | **0.86** |
| RandomForestAcc | 0.87 | 0.82 | 0.87 | 0.82 | **0.91** | **0.86** |
| OaxacaBlinder | 0.87 | 0.82 | 0.87 | 0.82 | 0.87 | 0.82 |
| HDPD (proposed) | **0.91** | **0.86** | **0.91** | **0.86** | **0.91** | **0.86** |

Table 4: Difference in AUCs between the revised insurance prediction model with respect to the top $k$ variables identified by the proposed versus `RandomForestAcc` procedures (Diff AUC-k = Proposed − `RandomForestAcc`). 95% CIs are shown.

| k | Diff AUC-k | Lower CI | Upper CI |
|---|---|---|---|
| 1 | 0.000 | 0.000 | 0.000 |
| 2 | **0.006** | 0.001 | 0.010 |
| 3 | -0.002 | -0.007 | 0.002 |
| 4 | 0.004 | -0.002 | 0.008 |
| 5 | -0.001 | -0.006 | 0.003 |
| 6 | -0.002 | -0.008 | 0.003 |
| 7 | **0.007** | 0.002 | 0.011 |
| 8 | **0.006** | 0.001 | 0.010 |

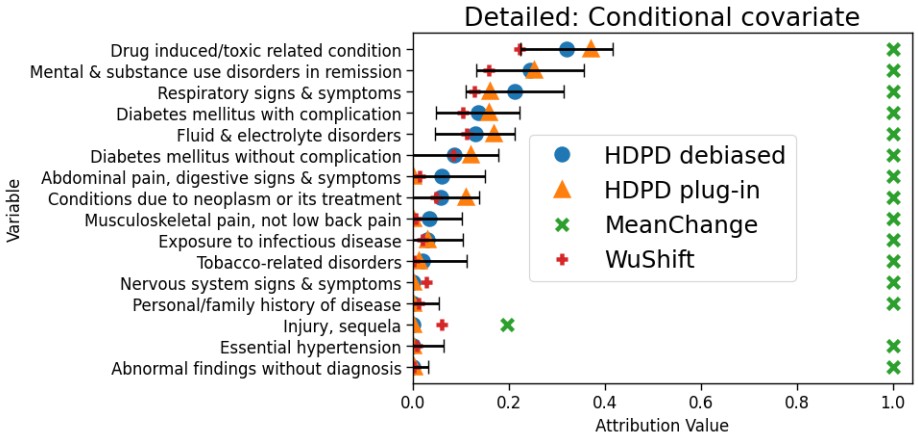

Figure 7: Detailed decompositions for the performance gap of a model predicting hospital readmission in HF population. Plot shows values for the full set of 16 variables.

codes ($Z$). While training, we reweigh samples by class weights to address class imbalance. Figure 7 shows the detailed decomposition of conditional covariate shift for the dataset.

**ACS Public Coverage.** We extract data from the American Community Survey (ACS) to predict whether a person has public health insurance. The data only contains persons of age less than 65 and having an income of less than $30,000. We analyze a neural network (MLP) trained on data from Nebraska (source) to data from Louisiana (target) given 3000 and 6000 observations from the source and target domains, respectively. Another 3300 from source for training the model. Figure 8 shows the detailed decomposition of conditional outcome shift for the dataset. Table 4 shows the difference in AUCs for the revised models based on top features from the proposed method versus `RandomForestAcc` method.


Figure 8: Detailed decompositions for the performance gap of a model predicting insurance coverage prediction across two US states (NE → LA). Plot shows values for the full set of 31 covariates.

- The abstract and/or introduction should clearly state the claims made, including the contributions made in the paper and important assumptions and limitations. A No or NA answer to this question will not be perceived well by the reviewers.
- The claims made should match theoretical and experimental results, and reflect how much the results can be expected to generalize to other settings.
- It is fine to include aspirational goals as motivation as long as it is clear that these goals are not attained by the paper.

2. **Limitations**

Question: Does the paper discuss the limitations of the work performed by the authors?

Answer: [Yes]

Justification: Limitations are discussed in Section 7.

Guidelines:

- The answer NA means that the paper has no limitation while the answer No means that the paper has limitations, but those are not discussed in the paper.
- The authors are encouraged to create a separate "Limitations" section in their paper.
- The paper should point out any strong assumptions and how robust the results are to violations of these assumptions (e.g., independence assumptions, noiseless settings, model well-specification, asymptotic approximations only holding locally). The authors should reflect on how these assumptions might be violated in practice and what the implications would be.
- The authors should reflect on the scope of the claims made, e.g., if the approach was only tested on a few datasets or with a few runs. In general, empirical results often depend on implicit assumptions, which should be articulated.
- The authors should reflect on the factors that influence the performance of the approach. For example, a facial recognition algorithm may perform poorly when image resolution is low or images are taken in low lighting. Or a speech-to-text system might not be

used reliably to provide closed captions for online lectures because it fails to handle technical jargon.

- The authors should discuss the computational efficiency of the proposed algorithms and how they scale with dataset size.
- If applicable, the authors should discuss possible limitations of their approach to address problems of privacy and fairness.
- While the authors might fear that complete honesty about limitations might be used by reviewers as grounds for rejection, a worse outcome might be that reviewers discover limitations that aren't acknowledged in the paper. The authors should use their best judgment and recognize that individual actions in favor of transparency play an important role in developing norms that preserve the integrity of the community. Reviewers will be specifically instructed to not penalize honesty concerning limitations.

3. **Theory Assumptions and Proofs**

Question: For each theoretical result, does the paper provide the full set of assumptions and a complete (and correct) proof?

Answer: [Yes]

Justification: Assumptions and proofs for the theoretical results are provided in Sections 3, F, and G.

Guidelines:

- The answer NA means that the paper does not include theoretical results.
- All the theorems, formulas, and proofs in the paper should be numbered and cross-referenced.
- All assumptions should be clearly stated or referenced in the statement of any theorems.
- The proofs can either appear in the main paper or the supplemental material, but if they appear in the supplemental material, the authors are encouraged to provide a short proof sketch to provide intuition.
- Inversely, any informal proof provided in the core of the paper should be complemented by formal proofs provided in appendix or supplemental material.
- Theorems and Lemmas that the proof relies upon should be properly referenced.

4. **Experimental Result Reproducibility**

Question: Does the paper fully disclose all the information needed to reproduce the main experimental results of the paper to the extent that it affects the main claims and/or conclusions of the paper (regardless of whether the code and data are provided or not)?

Answer: [Yes]

Justification: Pseudocode is provided in Algorithms 1, 2, 3, and 4. Implementation details are described in Section H.

Guidelines:

- The answer NA means that the paper does not include experiments.
- If the paper includes experiments, a No answer to this question will not be perceived well by the reviewers: Making the paper reproducible is important, regardless of whether the code and data are provided or not.
- If the contribution is a dataset and/or model, the authors should describe the steps taken to make their results reproducible or verifiable.
- Depending on the contribution, reproducibility can be accomplished in various ways. For example, if the contribution is a novel architecture, describing the architecture fully might suffice, or if the contribution is a specific model and empirical evaluation, it may be necessary to either make it possible for others to replicate the model with the same dataset, or provide access to the model. In general. releasing code and data is often one good way to accomplish this, but reproducibility can also be provided via detailed instructions for how to replicate the results, access to a hosted model (e.g., in the case of a large language model), releasing of a model checkpoint, or other means that are appropriate to the research performed.

- While NeurIPS does not require releasing code, the conference does require all submissions to provide some reasonable avenue for reproducibility, which may depend on the nature of the contribution. For example
  - (a) If the contribution is primarily a new algorithm, the paper should make it clear how to reproduce that algorithm.
  - (b) If the contribution is primarily a new model architecture, the paper should describe the architecture clearly and fully.
  - (c) If the contribution is a new model (e.g., a large language model), then there should either be a way to access this model for reproducing the results or a way to reproduce the model (e.g., with an open-source dataset or instructions for how to construct the dataset).
  - (d) We recognize that reproducibility may be tricky in some cases, in which case authors are welcome to describe the particular way they provide for reproducibility. In the case of closed-source models, it may be that access to the model is limited in some way (e.g., to registered users), but it should be possible for other researchers to have some path to reproducing or verifying the results.

5. **Open access to data and code**

   Question: Does the paper provide open access to the data and code, with sufficient instructions to faithfully reproduce the main experimental results, as described in supplemental material?

   Answer: [Yes]

   Justification: Code is available at the link `https://github.com/jjfeng/HDPD`.

   Guidelines:

   - The answer NA means that paper does not include experiments requiring code.
   - Please see the NeurIPS code and data submission guidelines (`https://nips.cc/public/guides/CodeSubmissionPolicy`) for more details.
   - While we encourage the release of code and data, we understand that this might not be possible, so "No" is an acceptable answer. Papers cannot be rejected simply for not including code, unless this is central to the contribution (e.g., for a new open-source benchmark).
   - The instructions should contain the exact command and environment needed to run to reproduce the results. See the NeurIPS code and data submission guidelines (`https://nips.cc/public/guides/CodeSubmissionPolicy`) for more details.
   - The authors should provide instructions on data access and preparation, including how to access the raw data, preprocessed data, intermediate data, and generated data, etc.
   - The authors should provide scripts to reproduce all experimental results for the new proposed method and baselines. If only a subset of experiments are reproducible, they should state which ones are omitted from the script and why.
   - At submission time, to preserve anonymity, the authors should release anonymized versions (if applicable).
   - Providing as much information as possible in supplemental material (appended to the paper) is recommended, but including URLs to data and code is permitted.

6. **Experimental Setting/Details**

   Question: Does the paper specify all the training and test details (e.g., data splits, hyperparameters, how they were chosen, type of optimizer, etc.) necessary to understand the results?

   Answer: [Yes]

   Justification: Implementation and simulation details are described in Section H and Section I. Full details are in the provided code.

   Guidelines:

   - The answer NA means that the paper does not include experiments.
   - The experimental setting should be presented in the core of the paper to a level of detail that is necessary to appreciate the results and make sense of them.

- The full details can be provided either with the code, in appendix, or as supplemental material.

7. **Experiment Statistical Significance**

   Question: Does the paper report error bars suitably and correctly defined or other appropriate information about the statistical significance of the experiments?

   Answer: [Yes]

   Justification: Results are accompanied by confidence intervals.

   Guidelines:

   - The answer NA means that the paper does not include experiments.
   - The authors should answer "Yes" if the results are accompanied by error bars, confidence intervals, or statistical significance tests, at least for the experiments that support the main claims of the paper.
   - The factors of variability that the error bars are capturing should be clearly stated (for example, train/test split, initialization, random drawing of some parameter, or overall run with given experimental conditions).
   - The method for calculating the error bars should be explained (closed form formula, call to a library function, bootstrap, etc.)
   - The assumptions made should be given (e.g., Normally distributed errors).
   - It should be clear whether the error bar is the standard deviation or the standard error of the mean.
   - It is OK to report 1-sigma error bars, but one should state it. The authors should preferably report a 2-sigma error bar than state that they have a 96% CI, if the hypothesis of Normality of errors is not verified.
   - For asymmetric distributions, the authors should be careful not to show in tables or figures symmetric error bars that would yield results that are out of range (e.g. negative error rates).
   - If error bars are reported in tables or plots, The authors should explain in the text how they were calculated and reference the corresponding figures or tables in the text.

8. **Experiments Compute Resources**

   Question: For each experiment, does the paper provide sufficient information on the computer resources (type of compute workers, memory, time of execution) needed to reproduce the experiments?

   Answer: [Yes]

   Justification: Compute resources are described in Section I.

   Guidelines:

   - The answer NA means that the paper does not include experiments.
   - The paper should indicate the type of compute workers CPU or GPU, internal cluster, or cloud provider, including relevant memory and storage.
   - The paper should provide the amount of compute required for each of the individual experimental runs as well as estimate the total compute.
   - The paper should disclose whether the full research project required more compute than the experiments reported in the paper (e.g., preliminary or failed experiments that didn't make it into the paper).

9. **Code Of Ethics**

   Question: Does the research conducted in the paper conform, in every respect, with the NeurIPS Code of Ethics `https://neurips.cc/public/EthicsGuidelines`?

   Answer: [Yes]

   Justification: We adhere to the NeurIPS Code of Ethics.

   Guidelines:

   - The answer NA means that the authors have not reviewed the NeurIPS Code of Ethics.

- If the authors answer No, they should explain the special circumstances that require a deviation from the Code of Ethics.
- The authors should make sure to preserve anonymity (e.g., if there is a special consideration due to laws or regulations in their jurisdiction).

10. **Broader Impacts**

Question: Does the paper discuss both potential positive societal impacts and negative societal impacts of the work performed?

Answer: [Yes]

Justification: Although we do not envision any direct negative impacts of the work, potential positive and negative impacts are discussed in Section B.

Guidelines:

- The answer NA means that there is no societal impact of the work performed.
- If the authors answer NA or No, they should explain why their work has no societal impact or why the paper does not address societal impact.
- Examples of negative societal impacts include potential malicious or unintended uses (e.g., disinformation, generating fake profiles, surveillance), fairness considerations (e.g., deployment of technologies that could make decisions that unfairly impact specific groups), privacy considerations, and security considerations.
- The conference expects that many papers will be foundational research and not tied to particular applications, let alone deployments. However, if there is a direct path to any negative applications, the authors should point it out. For example, it is legitimate to point out that an improvement in the quality of generative models could be used to generate deepfakes for disinformation. On the other hand, it is not needed to point out that a generic algorithm for optimizing neural networks could enable people to train models that generate Deepfakes faster.
- The authors should consider possible harms that could arise when the technology is being used as intended and functioning correctly, harms that could arise when the technology is being used as intended but gives incorrect results, and harms following from (intentional or unintentional) misuse of the technology.
- If there are negative societal impacts, the authors could also discuss possible mitigation strategies (e.g., gated release of models, providing defenses in addition to attacks, mechanisms for monitoring misuse, mechanisms to monitor how a system learns from feedback over time, improving the efficiency and accessibility of ML).

11. **Safeguards**

Question: Does the paper describe safeguards that have been put in place for responsible release of data or models that have a high risk for misuse (e.g., pretrained language models, image generators, or scraped datasets)?

Answer: [NA]

Justification: We do not release a model or a new dataset.

Guidelines:

- The answer NA means that the paper poses no such risks.
- Released models that have a high risk for misuse or dual-use should be released with necessary safeguards to allow for controlled use of the model, for example by requiring that users adhere to usage guidelines or restrictions to access the model or implementing safety filters.
- Datasets that have been scraped from the Internet could pose safety risks. The authors should describe how they avoided releasing unsafe images.
- We recognize that providing effective safeguards is challenging, and many papers do not require this, but we encourage authors to take this into account and make a best faith effort.

12. **Licenses for existing assets**

Question: Are the creators or original owners of assets (e.g., code, data, models), used in the paper, properly credited and are the license and terms of use explicitly mentioned and properly respected?

Answer: [Yes]

Justification: Medical dataset used for the case study is accessed and analyzed under the terms of an IRB.

Guidelines:

- The answer NA means that the paper does not use existing assets.
- The authors should cite the original paper that produced the code package or dataset.
- The authors should state which version of the asset is used and, if possible, include a URL.
- The name of the license (e.g., CC-BY 4.0) should be included for each asset.
- For scraped data from a particular source (e.g., website), the copyright and terms of service of that source should be provided.
- If assets are released, the license, copyright information, and terms of use in the package should be provided. For popular datasets, `paperswithcode.com/datasets` has curated licenses for some datasets. Their licensing guide can help determine the license of a dataset.
- For existing datasets that are re-packaged, both the original license and the license of the derived asset (if it has changed) should be provided.
- If this information is not available online, the authors are encouraged to reach out to the asset's creators.

13. **New Assets**

    Question: Are new assets introduced in the paper well documented and is the documentation provided alongside the assets?

    Answer: [NA]

    Justification: We do not introduce new assets. The code is provided only for reproducing the experiments.

    Guidelines:

    - The answer NA means that the paper does not release new assets.
    - Researchers should communicate the details of the dataset/code/model as part of their submissions via structured templates. This includes details about training, license, limitations, etc.
    - The paper should discuss whether and how consent was obtained from people whose asset is used.
    - At submission time, remember to anonymize your assets (if applicable). You can either create an anonymized URL or include an anonymized zip file.

14. **Crowdsourcing and Research with Human Subjects**

    Question: For crowdsourcing experiments and research with human subjects, does the paper include the full text of instructions given to participants and screenshots, if applicable, as well as details about compensation (if any)?

    Answer: [NA]

    Justification: Study does not involve crowdsourcing or research with human subjects.

    Guidelines:

    - The answer NA means that the paper does not involve crowdsourcing nor research with human subjects.
    - Including this information in the supplemental material is fine, but if the main contribution of the paper involves human subjects, then as much detail as possible should be included in the main paper.
    - According to the NeurIPS Code of Ethics, workers involved in data collection, curation, or other labor should be paid at least the minimum wage in the country of the data collector.

15. **Institutional Review Board (IRB) Approvals or Equivalent for Research with Human Subjects**

Question: Does the paper describe potential risks incurred by study participants, whether such risks were disclosed to the subjects, and whether Institutional Review Board (IRB) approvals (or an equivalent approval/review based on the requirements of your country or institution) were obtained?

Answer: [Yes]

Justification: Medical dataset was accessed according to the terms of use of an IRB.

Guidelines:

- The answer NA means that the paper does not involve crowdsourcing nor research with human subjects.
- Depending on the country in which research is conducted, IRB approval (or equivalent) may be required for any human subjects research. If you obtained IRB approval, you should clearly state this in the paper.
- We recognize that the procedures for this may vary significantly between institutions and locations, and we expect authors to adhere to the NeurIPS Code of Ethics and the guidelines for their institution.
- For initial submissions, do not include any information that would break anonymity (if applicable), such as the institution conducting the review.

