# OpenReview forum: "A hierarchical decomposition for explaining ML performance discrepancies"
_NeurIPS.cc/2024/Conference — NeurIPS 2024 poster_

### Official Review · Reviewer_7cwp · 2024-06-21

**Soundness:** 2
**Presentation:** 3
**Contribution:** 3
**Rating:** 6
**Confidence:** 3

**Summary:**

The paper proposes a hierarchical decomposition model for binary machine learning classifiers. The nonparametric model allows for a detailed decomposition of distributions shifts at aggregate or partial levels. Additionally, confidence intervals for the proposed estimators are presented.

**Strengths:**

1. Introducing a new general concept to decompose distribution shifts for classifiers has a wide range of applications. Specifically basing the evaluation on $R^2$-measures enhances interpretability.

2. While no causal graph is required, causal knowledge can easily be incorporated into the setting.

3. The paper is well-organized and gradually motivates the complex definitions.

**Weaknesses:**

1. The proposed method is restricted to (binary) classification problems. This limits the applicability and should already be noted in the abstract and/or title.

2. The framework requires baseline variables W, which can not be used for partial shifts. This is already sufficiently acknowledged and well motivated, especially combined with a causal graph.

3. The conditions should be discussed in more detail. Specifically, the fast convergence rate (8) is a very strong assumption that is not sufficiently discussed.

Minor Comments:
- Figure 2(a): No dashed lines as mentioned in the caption.
- L.162: Typo “toderive”
- L. 170: Typo “psuedocode”

**Questions:**

1. Regarding Weakness 3. The reference [2] only achieves fast convergence under strong and specific conditions. I am not quite sure whether these conditions are even achievable in the presented setting. Specifically, the binned $q_{\text{bin}}$ is not smooth, which would already violate the Hölder condition. Further, the margin condition should be discussed as this would require comparatively “simple” classification problems based on the bins.

**Limitations:**

See Weaknesses

---

> ### Author Rebuttal · Authors · 2024-08-07
>
> We thank the reviewer for a careful reading of the work. We are encouraged to hear that the reviewer found the manuscript well-organized and the method widely applicable.
>
> 1. **Binary classification**: While the manuscript focuses on binary classification, the framework can be readily extended to multi-class classification problems as well as to regression problems. The only section of this work that is specific to binary classification is importance estimation of $s$-partial outcome shifts with respect to the binned probabilities $\Pr(Y=1|Z,W) = E[Y|Z,W]$. However, one can just as easily define $s$-partial outcome shifts with respect to the binned expectations, which would make the framework applicable to regression and multi-class settings. We will add a section to the appendix to describe how such an extension can be achieved. Thank you for this suggestion!
>
> 2. **Detailed decomposition of $W$**: As the reviewer noted, the current framework does not provide a detailed decomposition of $W$ variables. We had previously considered simply rotating the roles of $W$ and $Z$, which would give us detailed decompositions for all variables. However, it was unclear how to interpret the resulting values. As such, we have left this open question for future work.
>
> 3. **Fast convergence rate assumption**: Thank you for the question on the fast convergence rate assumption! While we agree that fast convergence rates are not achievable in all settings, it does hold, for instance, when the set of $x=(w,z)$ with probabilities near the bin edges $b$ is measure zero. That is, super fast (exponential) convergence rates hold if the set $\Xi\_{\epsilon} = \left\\{x: \left|Q(x) - b\right| \le \epsilon \right\\}$ is zero for some $\epsilon > 0$, which is a relaxation of Condition 3.1 in the manuscript. We do not need $q_{\text{bin}}$ to satisfy the Hölder condition; instead, we just need $q$, the function prior to binning, to satisfy the Hölder condition. In fact, our convergence rates follow from reference [2] quite directly, as they similarly threshold the estimated probability function and demonstrate fast convergence of the binned function.
>
> We believe that this margin condition is quite reasonable in practice, as there may very well be some small $\epsilon$ for which the margin condition holds. More importantly, we have found that the method performs quite well, even under violations of this assumption. As shown in the empirical results, our method is able to achieve the nominal Type I error rates, even though the generated data does not satisfy the margin condition. We will revise the manuscript with this discussion of the fast convergence rate assumption.
>
> References:
>
> 2. Jean-Yves Audibert and Alexandre B Tsybakov. Fast learning rates for plug-in classifiers. Ann. Stat., 35(2):608–633, April 2007.

---

> > ### Comment · Reviewer_7cwp · 2024-08-09
> >
> > Thank you for addressing my comments.
> >
> > I think the paper should include a detailed discussion on conditions and examples which would satisfy the corresponding convergence properties as these might be quite restrictive.
> > Further, as fast convergence is required for $q_{\text{bin}}$ propensity and not $Q(x)$, I still have doubts about the rate assumptions. A detailed proof with precise statements would be helpful.

---

> > > ### Author Response · Authors · 2024-08-09
> > > **Response by Authors**
> > >
> > > Thank you for the suggestions. We will update the paper with a more detailed discussion on the conditions and provide examples.
> > >
> > > To address the reviewer's doubts, here is the precise theorem for the simple case with two bins, i.e. $Q_{bin} = 1\\{Q(x) \ge 0.5\\}$ where function $Q$ is the conditional probability of $Y=1$ given $x$. The result and proof are essentially identical to Proposition 3.7 in Audibert and Tsybakov 2007, except for some minor variations due to a slightly different loss function:
> > >
> > > **Theorem.** Let $Q_{bin}(x) = 1\\{Q(x) \ge 0.5\\}$. Suppose the function $Q$ belongs to the Hölder class $\Sigma(\beta, L, \mathbb{R}^d)$ for positive integer $\beta$ and $L > 0$ (see definition in Audibert and Tsybakov 2007),  the marginal law of $X$ satisfies the strong density assumption, and the margin condition $\Pr(0<|Q(X) - 1/2| \le t_0) =0$ is satisfied for some $t_0>0$. Then the rate of convergence of the plug-in classifier $\hat{Q}\_{bin,n}(x) = 1\\{\hat{Q}\_{n}(x) \ge 0.5\\}$ is exponential, i.e.
> > > $$
> > >     \mathbb{E}\left(
> > >     \hat{Q}\_{bin,n}(X) - Q\_{bin}(X)
> > >     \right)^2 \le C_1 \exp(-C_2 n)
> > > $$
> > > for constants $C_1, C_2 > 0$ that do not depend on $n$.
> > >
> > > *Proof.*
> > > Per the margin condition and Lemma 3.6 in Audibert and Tsybakov 2007, we have
> > > $$
> > > \mathbb{E}\left(\hat{Q}\_{bin,n}(X) - Q\_{bin}(X)\right)^2
> > > = \Pr\left(\hat{Q}\_{bin,n}(X) \ne Q\_{bin}(X)\right)
> > > \le \Pr\left(
> > > \left| \hat{Q}\_n(X) - Q(X)\right| > t_0
> > > \right).
> > > $$
> > > Then apply Theorem 3.2 (equation 3.7) in Audibert and Tsybakov 2007 with $\delta= t_0$.
> > >
> > > This result can be easily extended to the case with multiple bins, which we will include in the paper. We hope that this result clarifies that fast convergence for $Q_{bin}$ is achievable under appropriate conditions. We note that the condition that $Q$ is Hölder is not restrictive since it is satisfied whenever the conditional probability of $Y=1$ given $x$ is continuously differentiable in $x$ e.g. a logistic regression function.
> > >
> > > In light of these responses, we kindly ask the reviewer to consider raising their score.

---

> > > > ### Comment · Reviewer_7cwp · 2024-08-10
> > > >
> > > > Thank you for your response. I still have some follow-up questions.
> > > >
> > > >
> > > > Does this require that some type of margin condition will hold for any bin edge?
> > > > This would be considerably stronger than Condition 3.1. Additionally, combined with the Hölder conditions and the large number of bins (e.g. $B=20$) this would restrict the applicability of the results to a very few scenarios. A similar discussion as after Theorem 3.3. in Audibert and Tsybakov 2007 would be helpful.
> > > >
> > > > Further, Theorem 3.2. considers a local polynomial estimator and would not be generally valid for any type of plug-in estimate of $q(W,Z)$.
> > > >
> > > > I think a thorough discussion of the conditions that enable fast convergence rates of $q_{\text{bin}}$ will clarify the applicability of the paper.
> > > > Generally, I really like the concept and will update my rating.
> > > >
> > > > Minor comment:
> > > > $q_{\text{bin}}(w,z)$ creates $B+1$ bins

---

> > > > > ### Author Response · Authors · 2024-08-12
> > > > > **Response by Authors**
> > > > >
> > > > > Thank you for updating the rating! We will provide a thorough discussion of the conditions for fast convergence rates of $Q_{\text{bin}}$, including sufficient conditions and examples. As the reviewer rightly noted, in the case with multiple bins, a sufficient condition for fast convergence rates to hold is that the margin condition $\Pr(0<|Q(X) - c| \le t_0) =0$ holds for some $t_0>0$ for all bin edges $c$ (except $c \in\\{0,1\\}$). Requiring this condition to hold for $B=20$ bins is certainly stronger than requiring it to hold for $B=2$. Nevertheless, we found that 20 uniformly-sized bins worked well in both real-world datasets. We appreciate the thorough feedback from the reviewer and will update the paper to discuss the theoretical and practical implications of making such assumptions.

---

### Official Review · Reviewer_fcBN · 2024-06-28

**Soundness:** 3
**Presentation:** 1
**Contribution:** 4
**Rating:** 8
**Confidence:** 2

**Summary:**

The authors describe a novel method to detect root causes of distribution shift (performance variability) of ML classifiers across domains. The method estimates how much of the shift is due to covariate shift vs. outcome shift, and which input features contribute most to the said shifts.

**Strengths:**

Very significant topic and, as far as I can tell, original approach. Good Real-world case studies. Code is provided.

**Weaknesses:**

The paper is not well written because most NeurIPS audience (myself included) may not be able to understand the methods. For example, this
is almost incomprehensible (lines 76-80):

An aggregate shift substitutes a factor from the source domain in (1) with that from the target domain, i.e. we swap the factor from p0 to p1. A partial shift with respect to variable subset s (or an s-partial shift) shifts a factor from the source domain in (1) only with respect to variable subset s; we denote this by swapping a factor from p0 to ps.

I am guessing the authors did not have enough space to write elaborate explanation, or maybe reading it requires extensive background in this
particular area. Suggestion: write only the final equations and high-level explanation how it would be applied in ML practice; the rest of the derivation should be moved to Technical Supplement, and since there is no length constraint there, you can provide extensive and clear derivation of your result.

Bottom line: my best guess is this is a strong paper, but I have low confidence in my assessment due to the above.

**Questions:**

What is the connection between equation (2) and VI? Is (2) the
definition of VI?

What are E000, E100 etc. between lines 89 and 90?

In Fig. 3, two types of results are shown: HDPD debiased and HDPD
plug-in. If someone decides to use your method, which version do you
recommend?

**Limitations:**

The limitations are addressed adequately.

---

> ### Author Rebuttal · Authors · 2024-08-07
>
> We thank the reviewer for their time and positive feedback. We appreciate the constructive criticism to improve the clarity of the paper, and have included a high-level description of the procedure in the global response. We will also revise the manuscript to focus more on the high-level explanation.
>
> * **Definition of VI**: Yes, the Shapley value formula in equation (2) defines the variable importance.
>
> * **Notation E000**: $\mathbb{E}\_{D\_W, D\_Y, D\_Y}$ denotes an expectation with respect to the joint distribution $p\_{D\_W}(W) p\_{D\_Z}(Z|W) p\_{D\_Y}(Y|W,Z)$. Thus $\mathbb{E}\_{000}$ and $\mathbb{E}\_{111}$  denote the expectations with respect to the source and target environments, respectively.
>
> * **Which version of method to use**: We generally recommend using the HDPD debiased method because it provides valid uncertainty quantification. In particular, its confidence intervals have valid coverage, asymptotically.

---

> > ### Comment · Reviewer_fcBN · 2024-08-10
> >
> > Thanks for the response. The high-level description that you provided is very helpful. Accordingly, I increased my level of Confidence to 2.

---

> > > ### Author Response · Authors · 2024-08-10
> > >
> > > We appreciate you taking the time to read our response and are happy to hear that it is helpful. Thank you!

---

> > > > ### Comment · Reviewer_fcBN · 2024-08-12
> > > >
> > > > To clarify, no more questions.

---

### Official Review · Reviewer_tcta · 2024-07-11

**Soundness:** 3
**Presentation:** 3
**Contribution:** 2
**Rating:** 5
**Confidence:** 4

**Summary:**

The problem of explaining and remedying a dropoff in the accuracy of a supervised learning model when applied to a new environment (i.e. a new joint distribution between inputs and output) is studied. The paper tackles these problems for a non-parametric setting and with the goal of identifying specific individual input variables which most contribute to the dropoff. Confidence intervals are also generated. Previous work in the area does not meet all these criteria. Input variables are segmented into two disjoint sets: a set of "baseline" variables W and a set of "conditional covariates" Z which are often (but not necessarily) causally downstream of the baseline variables. The total dropoff in performance is decomposed into the dropoff attributable to each of the 3 shifts: a shift in P(W), a shift in P(Z | W) and a shift in the conditional expectation of the target variable given the inputs, i.e., P(Y | Z; W).  The method is able to evaluate the impact of partial shifts in the Z variables, i.e. shifts in a subset of the Z variables. Partial shifts in W cannot be evaluated, though. To rank the impacts of shifts in individual Zs, Shapley values are used. To overcome nonparametric estimation challenges, a binning procedure as well as a Shapley value sampling procedure is used. The method is shown to give intuitive results for simulated data. Two real world problems are studied. The first problem is hospital readmission prediction. The shift in environment is from a general patient population to a patient population restricted to those with heart failure.  The second problem is health insurance coverage prediction where the shift in environment is from Nebraska to Louisiana. The method is evaluated in terms of the quality of the selection of the top-k variables most responsible for performance dropoff. The quality of the selection is evaluated by training and testing a model on the second environment where the inputs include the output of the original-environment variable, the baseline variables and only those top-k of the Z, rather than all the Z.  Superior results as compared to competing techniques are obtained.

**Strengths:**

The paper is original to the best of my knowledge.

If the flaws in the paper could be fixed, I certainly agree that the contribution of the work could be significant. Understanding and adapting to changing environments for tabular models is definitely an important problem and it's valuable to do this for general ML models rather than just linear stuff as in Oaxaca-Blinder.

Although I didn't have time to check every technical detail super-carefully, the paper does seem to overcome major technical challenges in a clever way. In particular, I quite liked the use of the random subsampling Shapley value approach to overcome the combinatorial explosion of exact Shapley value computation. I've wondered vaguely in the past whether a technique like that could be valid and I feel I benefitted from learning about that reference and seeing its application here.

Some parts of the paper are well written, although I also think there are serious clarity problems, as I'll explain in the next section.

**Weaknesses:**

I'm sure it's unpleasant for the authors to see my score of 3. I probably have never liked a paper I've given a 3 nearly as much as I liked this paper. I think the potential is here to achieve (eventually) a quite significant contribution to the ML literature. However, I feel strongly that the paper is not ready for submission as is.  I'm somewhat open to raising my score if my concerns can be addressed in rebuttal, but it woh't be easy to address my concerns given the short rebuttal time window.

The biggest flaw I see in the paper is that the distinction between baseline variables W and conditional covariates Z is very problematic. The distinction is not well explained and it's poorly illustrated both by the real-world problems and the synthetic experiments. The authors say that Z can often be thought of as causally downstream of W but this not required. Ok, if it's not required, then how else would you segment the variables into W and Z ? I don't think this is ever explained.  For the real world problems (unless I missed something), there doesn't seem to be any explanation why some variables are W and others are Z. For hospital readmission, why would demographic variables be W and diagnosis codes be Z ? I get that demographics could cause the diagnoses to some extent (although the diagnoses are surely a long way from being fully caused by a small set of demographic variables) but I still needed a lot more explanation here. For ACS insurance, why are the 3 demographic variables (age, race etc) the baseline Ws and the others (marital status, etc) the conditional covariates?  For the synthetic stuff, on line 243, why are W, Z1, Z2, Z3 independent normal if we are usually supposed to think of Z as causally downstream of W ?

I also consider only 2 real world problems to be a bit thin in terms of real-world results for NeurIPs acceptance.

Another problem I have with the experiments is that more naive  baselines are needed to double-check that the partial retraining beats the baseline. I'll take the authors' word that this procedure of taking the old-environment model output as a new-environment input and combining that with top-k most impactful Zs is some sort of standard in the literature. Nonetheless, I think you should compare it against:

1) Throw away all the old-environment data and fully retrain a new-environment-training-data-only model with all available W and Z inputs.

2) Do some sort of weighted-data retraiing using all inputs, where old environment and new environment training data are both used by old environment is weighted less (with the weight being a CV-optimized hyperparameter).

There are also some clarity problems.

On line 147, it's wrong to use "risk" as the probability of class 1. "Risk" has an established meaning in the ML community as "loss" or "error". E.g. "empirical risk minimization" means loss minimization, not minimization of the odds of class 1.

The paper also makes various variable/terminology choices which aren't a big problem but could be improved so that the reader finds it easier to follow.

I would suggest using "environment" where "domain" is used because "domain" can also mean the input space, i.e. a function mapping from domain to range, so that could be confusing.

I would not call the 2nd domain/environment "target", since that can be confused with Y. Maybe the "shifted" environment or the "new" environment ?

I wouldn't denote the two domains/environments as 0 and 1. I would use letters, e.g. O and S for original and shifted.


Typos:

L94 which have the benefits-> which has the benefits
L162 toderive

**Questions:**

How does one segment between W and Z if not by casual knowledge (since this is not required) ?

You need at least one W for your method. Can you just arbitrarily pick 1 variable for your W ? Is there some sort of bias/variance or other tradeoff if you picked 1 variable as W and e.g. the other 99 as Z or would 20/80 or 50/50 be more reasonable for some reason?

Why do you think the W you chose cause the Z you chose in hospital readmission and insurance coverage?

** update after rebuttal **

In light of the additional explanations for how to distinguish between W and Z and in light of the additional experiments evaluating the baselines I suggested, I raised my score to a 5.

**Limitations:**

Adequately addressed.

---

> ### Author Rebuttal · Authors · 2024-08-07
>
> We thank the reviewer for a thorough reading of the work and the constructive feedback. We are excited to hear that you liked the work and hope that our response will convince you to raise the score.
>
> * **Selecting $W$ and $Z$**: Thank you for asking this question on how to partition variables into $W$ and $Z$! Your question has prompted us to think more deeply about the selection procedure, and we sincerely thank the reviewer for the opportunity to clarify our thinking in this area. We have responded in the global response with both causal and non-causal approaches for choosing $W$ versus $Z$. We believe these options provide users many practical approaches to applying our method.
>
> * **Choice of $W$ and $Z$ in experiments**: Our reasoning was as follows:
>    + **Hospital readmission**: The primary reason for setting demographic variables to $W$ and diagnoses to $Z$ is that demographics are causally upstream of diagnoses, and diagnoses are causally upstream of readmission status $Y$. As the reviewer rightly noted, other variables can also cause diagnosis codes. However, this is not an issue as long as we correctly interpret the results. In particular, Figure 3(a) reports how well shifts in disease diagnoses explain the variation in the performance gaps (between the two environments) across demographic groups $W$. Omitting variables from $W$ simply changes the performance gaps we are trying to explain (see Subbaswamy et al. 2021 for a formal discussion of why there is a causal interpretation of distribution shifts with respect to $W$, even if it does not contain all possible confounders). At the same time, because one is unlikely to include all possible variables that cause $Y=$readmission in $Z$, the estimated variable importances should be viewed as *relative* to variables included in $Z$ and not as *absolute* importances with respect to all possible variables. We will discuss this nuance in more detail in the manuscript.
>    + **ACS insurance**: Most of the variables available in the ACS dataset could be considered as demographics. Nevertheless, the selected $W$ (sex, age, race) are still causally upstream of the remaining variables we assigned to $Z$ (health condition, marital status, education, etc). One can then interpret the explanations similar to the aforementioned example. Alternatively, one can think of this from an ML fairness perspective, where our goal is to explain the performance gap overall as well as performance gaps within groups defined by the protected attributes of sex, age, and race. Again, as the causal graph may be missing variables, the results should be interpreted per the discussion above.
>    + **Simulation setup described on line 243**: This simulation tests our explanation method for conditional outcome shifts. Because the outcome distribution $Y|W, Z$ is (variationally) independent of the conditional distribution of $Z|W$, both the estimand and the estimation procedure are not affected by the dependence structure between $Z$ and $W$. As such, we considered a simple simulation where $Z$ and $W$ are completely independent. Nevertheless, we see how this can be confusing to the reader and will revise the manuscript so that $Z$ depends on $W$.
>
> * **Bias-variance tradeoff between $W$ and $Z$**: This is an interesting question! When more variables are assigned to $W$, the performance gap with respect to $W$ (i.e. $\Delta_{\cdot 10}(W)$) is a more complex function. Thus we may have more uncertainty in our estimate of $\Delta_{\cdot 10}(W)$, which may lead to wider confidence intervals for the variable importance values. On the other hand, more variables in $W$ lead to higher variance of $\Delta_{\cdot 10}(W)$, so it allows one to better distinguish the relative importance of variables in $Z$ for explaining its variability. We will include this nuance in the manuscript as an additional consideration when choosing $W$ and $Z$.
>
> * **Partial retraining experiments**: We have included the two suggested baselines from the reviewer in the table below. Partial retraining based on the proposed framework is better than the two baselines: it results in models with a 2\% improvement in AUC. This is because targeted retraining is more statistically efficient (i.e. requires less training data), so it can quickly adapt to the new environment.
>
> * We thank the reviewer for the suggested terminology changes such as risk to conditional outcome, domain to environment, and target to shifted environment. We will incorporate these into our revision.
>
> **Table**: AUC and accuracy of retrained models are reported. Target-only model is retrained with all features ($W,Z$) but only on the target data. Weighted source-target model is retrained on pooled source and target data with all features, where source data are assigned weight $r$ to optimize AUC in the target environment. Retrained models based on the proposed method achieve the best performance.
> | Method                               | AUC      | Accuracy |
> |--------------------------------------|----------|----------|
> | ParametricChange                     | 0.87     | 0.82     |
> | ParametricAcc                        | 0.87     | 0.82     |
> | RandomForestAcc                      | 0.87     | 0.82     |
> | OaxacaBlinder                        | 0.87     | 0.82     |
> | HDPD (proposed)                      | **0.91** | **0.86** |
> | *(new) Target-only model*            | 0.89     | 0.84     |
> | *(new) Weighted source-target model* | 0.89     | 0.85     |
>
> References:
>
> Subbaswamy, Adarsh, Roy Adams, and Suchi Saria. “Evaluating Model Robustness and Stability to Dataset Shift.” International Conference on Artificial Intelligence and Statistics 2021 https://proceedings.mlr.press/v130/subbaswamy21a.html.

---

> > ### Comment · Reviewer_tcta · 2024-08-09
> > **Raised my score to a 5**
> >
> > I have a better understanding of how to choose W vs Z now; thank you.
> >
> > I also appreciate the additional experiments.

---

> > > ### Author Response · Authors · 2024-08-10
> > >
> > > We appreciate you taking the time to read our response and are happy to hear that it is helpful. Thank you!

---

### Official Review · Reviewer_tcwU · 2024-07-12

**Soundness:** 3
**Presentation:** 2
**Contribution:** 3
**Rating:** 5
**Confidence:** 2

**Summary:**

This paper introduces a hierarchical decomposition framework aimed at explaining performance discrepancies of machine learning models across different domains. It proposes both aggregate and detailed decompositions to quantify the impact of shifts in feature distributions (marginal and conditional) on model performance, without necessitating detailed causal graphs. The framework leverages nonparametric methods and Shapley-based variable attributions, enhancing understanding and facilitating targeted interventions to mitigate performance gaps. Coverage rates and further experiment analysis are provided.

**Strengths:**

The scope of the paper significantly extends that of the most relevant prior work, which provides the aggregate decomposition that this work builds on.

 This work further extends several other related works by characterizing shifts that are more complex and in a setting where no causal graph encoding prior knowledge of the data generating process or shift structure is assumed.

The estimation and inference procedures are provided, with convergence rates.

**Weaknesses:**

I find the paper, at times, is difficult to follow. I would suggest a formal definition on the framework, along on definition of performance gap, and each level of explanation. Some causal interpretation in appendix could be useful in Section 2 to understand the framework.

Experiment results bring some questions.

**Questions:**

- L85 aggregate: the first level mentioned here represents the aggregated level?
- linear decomposition: although it looks like just a log operation from equation, it would be good to give intuition on such a linear decomposition. What assumptions are made exactly here?
- More importantly, the performance gap may be from  a combination of variables, instead of one variable at a time. How does the proposed framework address such cases?
- How scalable is the proposed method? Would the proposed method handle discrete variables?
- Fig 3b: some variables seems to be highly correlated or even negation of each other ( e.g. Married, Divorced, Never married, and e.g. citizenship status). Yet they do not seems to be consistent to each other (e.g., married and divorced). What is the reason for such a difference?

**Limitations:**

yes

---

> ### Author Rebuttal · Authors · 2024-08-07
>
> We thank the reviewer for helpful comments and are glad that they appreciated the novel aspects of the work. We summarize the framework in the global response for more clarity and respond to the reviewer's questions below.
>
> * **Aggregate decomposition**: Each term in the aggregate decomposition $\Delta = \Lambda_W + \Lambda_Z + \Lambda_Y$ quantifies the performance change when we vary only one factor of the joint distribution of $(W,Z,Y)$.
> They are defined as:
> $$\Lambda_W = \mathbb{E}\_{100}[\ell] - \mathbb{E}\_{000}[\ell] = \int \ell(f(w,z), y) p_0(y|w,z) p_0(z|w) \left(p_1(w) - p_0(w)\right) dy dz dw$$
> $$\Lambda_Z = \mathbb{E}\_{110}[\ell] - \mathbb{E}\_{100}[\ell] = \int \ell(f(w,z), y) p_0(y|w,z) \left(p_1(z|w) - p_0(z|w)\right) p_1(w) dy dz dw$$
> $$\Lambda_Y = \mathbb{E}\_{111}[\ell] - \mathbb{E}\_{110}[\ell] = \int \ell(f(w,z), y) \left(p_1(y|w,z) - p_0(y|w,z)\right) p_1(z|w) p_1(w) dy dz dw$$
> where $\Lambda_W$ corresponds to the effect of varying $p(W)$, $\Lambda_Z$ corresponds to the additional effect of varying $p(Z|W)$, and $\Lambda_Y$ corresponds to the additional effect of varying $p(Y|Z,W)$.
> From the equations, one can see that the terms sum up to $\Delta$ by definition; no assumptions are needed.
>
> * **Multiple variables**: As the reviewer rightly noted, multiple variables can shift to induce a performance gap. This, in fact, is one of the major contributions of this work, as it provides a framework that (i) formally defines an $s$-partial shift for variable subsets $s$ and a value function $v(s)$ that quantifies how well a partial shift with respect to subset $s$ explains the performance gap and (ii) a procedure for estimation and statistical inference of the value function. Shapley values should be viewed as an abstraction that lies on top. Its primary use is to provide a simple digestible summary at the level of individual variables. Shapley values also account for interactions between variables, as it defines the importance of variable $j$ as the average increase in the value function when a variable $j$ is added to a subset $s$. Our framework allows one to directly analyze the importance of variable subsets *or* Shapley values of individual variables.
>
> * **Scability**: The method is highly scalable. Scalability is achieved in two ways.
> First, we use a subset sampling procedure to estimate Shapley values that maintains statistical validity. This allows us to compute hundreds of Shapley values.
> Second, one can group together variables and use the framework to quantify the importance of variable groups (rather than individual variables).
> For instance, variables may be grouped based on prior knowledge (e.g. genes in the same pathway) and a clustering procedure (e.g. grouping together variables that are highly correlated).
>
> * **Discrete**: The method handles discrete variables, as illustrated in the real-world experiments section. In the submission, we obtain Shapley values for binary variables as well as categorical variables.
> For categorical variables, we can either obtain importance for each category (e.g. report importance of different marital statuses) or group together all the categories to report a single importance score (e.g. report importance of marital status as a whole). We opted for the former but our procedure can also do the latter.
>
> * **Fig 3b question**: For binary variables, the framework reports a single importance; there is no consistency problem. For categorical variables, the importance of different values may not necessarily be similar, which should be expected. A shift with respect to a specific category is not directly connected to a shift with respect to a different category. This is why the different categories of marital status have different values; marital status has five categories in the ACS dataset.
>
> * We appreciate the suggestion to surface the causal interpretation in the main paper rather than the appendix. Indeed, our explanation framework can be interpreted from the view of stochastic interventions, assuming causal sufficiency, positivity, and SUTVA hold.

---

> > ### Author Response · Authors · 2024-08-13
> >
> > Thank you again for reading our work and sharing feedback! We wanted to remind the reviewer that the discussion period ends today (AoE). We hope that the global response helps clarify the framework, and the point-by-point responses help address specific questions. In light of our responses, we kindly ask the reviewer to consider raising their score.

---

### Author Rebuttal · Authors · 2024-08-07

We sincerely thank the reviewers for their constructive feedback and the positive response.

To recap, the aim of the paper is to address a major **methodological gap**: there are currently no nonparametric methods that provide a detailed explanation for why the performance of an ML algorithm differs between two environments, without assuming the detailed causal graph is known. Due to this vacuum, applied ML papers often contain a long list of factors that differ between two environments as the set of potential causes; however, such lists contain many false positives, as variables with large shifts may not necessarily contribute to the performance drop, and miss true positives, as variables with small shifts can still have a large impact on algorithm performance.

There is a **significant need for explanation methods**, as the lack of generalizability of ML algorithms presents major problems in the **safety** of ML algorithms (particularly in clinical settings) and is a major barrier to widespread adoption. Having explanations is crucial to ML consumers for taking steps to mitigate performance drops, regulators for outlining environments where the algorithm can be safely deployed in, and ML developers for designing methods that are more robust and generalizable.

Our proposed framework aims to address this methodological gap. We are heartened by the reviewers agreeing that this is **"a very significant topic" [fcBN]** and the method has the potential to make **"a quite significant contribution to the ML literature" [tcta]**. Moreover, all reviewers acknowledge that the method **"significantly extends prior work" [tcwU]** and **"overcomes major technical challenges" [tcta]**.

Reviewers asked for more clarity on the framework at a high-level, which we add here. The proposed hierarchical framework decomposes the performance gap (i.e. the difference in the expected loss $\Delta = \mathbb{E}_1[\ell] - \mathbb{E}_0[\ell]$) of a model between two environments. We suppose variables $X$ are partitioned into baseline variables $W$ and conditional covariates $Z$.

We suggest selecting $Z$ to be the variables that may act as mediators of the environment shift and/or variables whose associations with $Y$ are likely to be modified by the environment shift (i.e. effect modification). This selection can be chosen based on a high-level causal graph, where $W$ are variables known to be upstream of $Z$. For instance, if $Z$ are treatment variables and $W$ are baseline variables, one has the nice interpretation that a covariate shifts is a change in the treatment policy and an outcome shift is a change in the treatment effect across the two environments.

In the absence of *any* prior knowledge, another option is to choose $W$ as the variables for which one would like the expected loss given $W$ to be invariant across the two environments; this can be useful to promote fairness of ML algorithms across environments.
When this invariance does not hold, the proposed framework explains how variables $Z$ contribute to these differences, which can inform efforts to eliminate performance gaps. This last option is similar to how variables are typically chosen in disparity analyses [Jackson 2021]. For instance, to understand why income differs between males and females controlling for age, one would set domain to $D=$ gender, $W =$ age, and $Z$ as variables that may explain this disparity (e.g. marital status, employment status).

The **aggregate level** decomposes the performance gap $\Delta$ into $\Lambda_W + \Lambda_Z + \Lambda_Y$, which are the performance gaps when only one factor of the joint distribution of the aggregate variables $(W, Z, Y)$ is varied one at a time.
The three terms are (1) the shift in the expected loss due to a shift in the distribution of $W$ from source to target ($\Lambda_{W}$, baseline covariate shift), (2) the shift due to a shift in the distribution of $Z|W$ from source to target ($\Lambda_{Z}$, conditional covariate shift), and (3) the shift due to a shift in the distribution of $Y|Z,W$ from source to target ($\Lambda_{Y}$, outcome shift). For large terms in the aggregate decomposition, we can drill down deeper to get a more detailed understanding of which variables are most important.

A **detailed decomposition** of $\Lambda_Z$ quantifies the importance of variable $Z_j$ for explaining the variation in expected loss differences (performance gaps) due to aggregate conditional covariate shifts given $W$.
We do so by first defining an $R^2$ measure that quantifies the percent variability explained by *partial* shifts in the conditional covariate distribution with respect to variable subsets $s$.
For interpretability, we then summarize the importance of variable $Z_j$ using Shapley values, which quantify how the addition of variable $Z_j$ to variable subsets $s$ increases the $R^2$ value, on average.
The **detailed decomposition** of $\Lambda_Y$ is defined similarly, where the importance of variable $Z_j$ is how well it explains the variation in the expected loss differences due to aggregate outcome shifts given $W,Z$.

This framework is the first of its kind that **satisfies all of the following desiderata** crucial to being useful in real-world scenarios: (i) it does *not* require knowledge of the detailed causal graph, (ii) is non-parametric, and (iii) quantifies uncertainty in explanations through confidence intervals.

References:

Jackson John W. Meaningful Causal Decompositions in Health Equity Research: Definition, Identification, and Estimation Through a Weighting Framework. Epidemiology 2021.

---

### Decision · Program_Chairs · 2024-09-25

**Decision:**

Accept (poster)

**Comment:**

The paper solves an interesting and important problem: namely understanding how the performance of a trained model may diminish when it is deployed in a new environment. As pointed out by reviewers 7cwp and tcta, I strongly recommend that the authors add some clear, practical guidance about selecting W & Z and to include some details about the rate assumptions (as discussed in their back-and-forth with 7cwp) when revising this work.